# Survival and rapid resuscitation permit limited productivity in desert microbial communities

Stefanie Imminger [1,2,7], Dimitri V. Meier[1,6,7], Arno Schintlmeister [1,3], Anton Legin [4], Jörg Schnecker [1], Andreas Richter [1], Osnat Gillor [5], Stephanie A. Eichorst [1] & Dagmar Woebken [1] ✉

Microbial activity in drylands tends to be confined to rare and short periods of rain. Rapid growth should be key to the maintenance of ecosystem processes in such narrow activity windows, if desiccation and rehydration cause widespread cell death due to osmotic stress. Here, simulating rain with $^2H_2O$ followed by single-cell NanoSIMS, we show that biocrust microbial communities in the Negev Desert are characterized by limited productivity, with median replication times of 6 to 19 days and restricted number of days allowing growth. Genome-resolved metatranscriptomics reveals that nearly all microbial populations resuscitate within minutes after simulated rain, independent of taxonomy, and invest their activity into repair and energy generation. Together, our data reveal a community that makes optimal use of short activity phases by fast and universal resuscitation enabling the maintenance of key ecosystem functions. We conclude that desert biocrust communities are highly adapted to surviving rapid changes in soil moisture and solute concentrations, resulting in high persistence that balances limited productivity.

Drylands constitute 46% of the terrestrial surface[1,2], play an important role in the global carbon cycle[3] and are currently expanding due to climatic changes[4,5]. They are characterized by long periods of drought, with rare and short rain events, that last only a few days a year[6–9]. As the activity of desert microorganisms is largely confined to these short and unpredictable episodes of rain[10], so are main microbial-mediated ecosystem processes in desert soils[11]. However, it remains unclear how microbial functions are executed if a considerable proportion of desert soil microorganisms remain dormant after rehydration, or if they exhibit considerable delay in their response as reported in other semiarid soils[12,13]. Key information, such as the fraction and identity of

responding desert soil microorganisms and their resuscitation speed, is lacking. Further, information on desert soil microbial productivity is lacking, despite its importance in being linked to the dryland's contribution to the global carbon cycle[3]. It has been suggested that drydown and/or rapid rehydration of soils via rain can cause cell death[14], which in turn would necessitate rapid growth in the short window of rain-induced activity for maintaining desert soil microbial communities. However, in situ growth rate data from desert soil microorganisms that support this assumption are lacking.

Long-term persistence, facilitated by dormancy, is critical in order for desert microorganisms to survive extended droughts and thus

[1]Centre for Microbiology and Environmental Systems Science, Department of Microbiology and Ecosystem Science, University of Vienna, Vienna, Austria. [2]University of Vienna, Doctoral School in Microbiology and Environmental Science, Vienna, Austria. [3]Large-Instrument Facility for Environmental and Isotope Mass Spectrometry, Centre for Microbiology and Environmental Systems Science, University of Vienna, Vienna, Austria. [4]Faculty of Chemistry, Institute of Inorganic Chemistry, University of Vienna, Vienna, Austria. [5]Zuckerberg Institute for Water Research, Blaustein Institutes for Desert Research, Ben Gurion University of the Negev, Midreshet Ben Gurion, Israel. [6]Present address: Department of Ecological Microbiology, Bayreuth Center of Ecology and Environmental Research (BayCEER), University of Bayreuth, Bayreuth, Germany. [7]These authors contributed equally: Stefanie Imminger, Dimitri V. Meier. ✉e-mail: dagmar.woebken@univie.ac.at

remain members of the microbial community[15,16]. However, dormancy mechanisms of desert soil microorganisms remain elusive, as information on desiccation survival strategies is primarily based on cultures, mainly not stemming from desert soils[17,18]. Furthermore, spore formation, which is assumed to be a major survival mode of soil microorganisms, is not a common strategy in bacteria found in desert ecosystems[19–21]. A dormant state cannot be sustained indefinitely, as cells will sustain damage to their DNA, proteins, and membranes, and if this damage extends beyond the point of repair, it will ultimately lead to cell death[22]. Therefore, resuscitation is critical as it provides the opportunity to revive, repair, and prepare for the next phase of dormancy. Until now, analyses on microbial resuscitation in desert soils have focused only on changes in the community composition after rain[23,24], or targeted biocrust cyanobacteria[25,26]. Information on the remaining majority of the diverse microbial community, the physiologies employed when transitioning out of a dormant state and potential dependencies among community members during resuscitation are lacking.

We addressed these open questions in biological soil crusts (biocrusts) from the Negev Desert, Israel. Biocrusts cover ~30% of all dryland soils and thus ~12% of the global terrestrial surface[27]. They play important roles in nutrient[28] and trace gas cycling[29] and in preventing soil erosion[30–32]. Biocrusts are a suitable model system for these investigations, as they are devoid of plants and harbor high microbial biomass[21] of moderately diverse microbial communities composed of cyanobacteria, various hetero-, mixo- and autotrophic bacteria, as well as archaea, fungi and microalgae[20,33,34]. We investigated the proportion of cells that reactivated and their associated growth rates in simulated rain events using heavy water ($^2H_2O$), followed by single-cell nano-scale secondary ion mass-spectrometry (NanoSIMS) detecting the incorporated $^2H$. Resuscitated microbial populations, their resuscitation speed and the employed physiological processes were elucidated by genome-resolved metatranscriptomics in a highly resolved time series.

In this study, we document immediate and simultaneous resuscitation of the microbial community in rehydrated biocrusts independent of taxonomy and physiology, resolving how these communities can execute major microbial-driven processes in short activity periods. However, this collective activity is accompanied by limited growth, which could not maintain the biocrust microbial communities if short rain episodes would have caused major cell loss. Thus, we argue that desert biocrust microorganisms are well prepared for sudden hydration and desiccation, preventing cell mortality. Further, this microbial community is characterized by low production, which has important implication in biocrust preservation and restoration considering that biocrusts are critical in desert soil stabilization.

## Results

### Majority of biocrust cells resuscitate in a simulated rain event with slow growth rates

Deuterium ($^2H$) cellular isotope enrichments revealed that in almost all biocrust cells that were retrieved with the applied cell-separation and -concentration approach, anabolic pathways were reactivated by rain to a level sufficient for biomass production. For this analysis, Negev Desert biocrusts were exposed to a simulated rain event (~26% water content, corresponding to 75% water holding capacity) with heavy water (30% $^2H_2O$), resulting in wet crusts for up to 24 h (Fig. 1). This corresponds to the duration of the most frequently occurring rain events observed in the central Negev Desert (Supplementary Table 1). When reactivated with heavy water, microorganisms can covalently bind $^2H$, mainly in C-H bonds during de novo lipid synthesis via NADPH, and thereby incorporate the isotope tracer. Hydrogen isotope compositions of biocrust filamentous cyanobacteria and non-cyanobacterial single cells were determined by NanoSIMS, resulting in $^2H$ content data of individual cells (Fig. 2a–c and Supplementary Fig. 1). Within the first 3 h of hydration, 68.4% of the single cells were significantly enriched in $^2H$ and thus considered active (see "Methods"), reaching 91.0% after 12 h and 94.6 % after 24 h (all $p$ value < 0.00135) (Fig. 2d). All analyzed filamentous cyanobacteria were anabolically active after a 3-h hydration period ($p$ value < 0.00135) (Fig. 2d). The $^2H$ content continuously increased with the hydration time, yielding 0.60 at% (median) within non-cyanobacterial single cells after 24 h of incubation and 4.60 at% (median) within filamentous

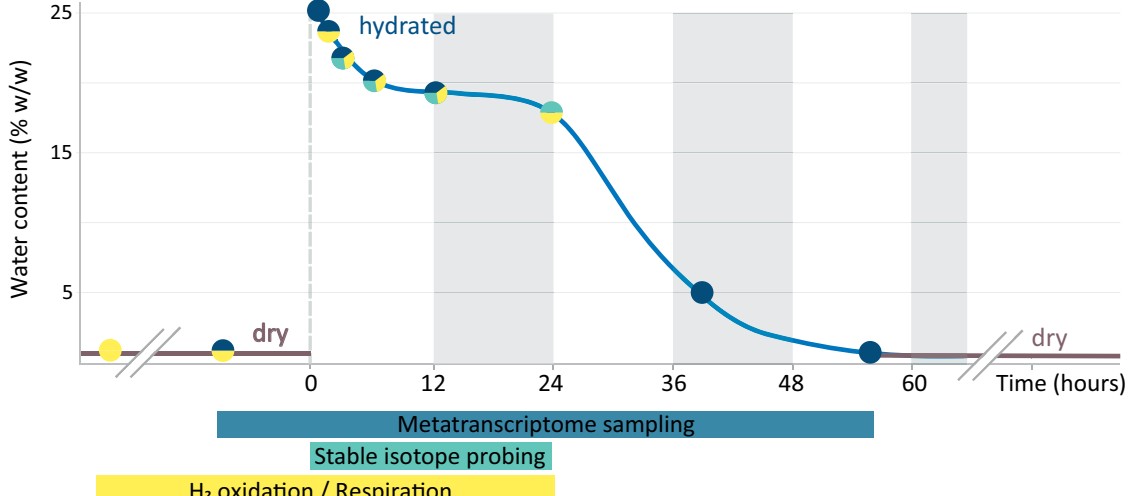

**Fig. 1 | Overview of assays applied during a simulated rain event consisting of biocrust hydration and subsequent desiccation.** The scheme depicts sampling time points for different experiments during the hydration/desiccation cycle under controlled day and night conditions in a climate chamber. The sampling time points were defined based on preliminary experiments. The biocrusts were hydrated to 75% of their water holding capacity corresponding to 26% water content/wet weight. Overall, eight time points were sampled in five replicates for the metatranscriptome investigations (blue). Triplicates were selected for sequencing based on their water content. For NanoSIMS analysis, biocrusts were incubated in a parallel experiment with 30% deuterium oxide (heavy water) and destructively sampled at four incubation times spanning the first 24 h after hydration (green). To monitor $H_2$ oxidation and respiration in dry and hydrated conditions, biocrust samples were incubated in triplicates over a period of several months for the dry and up to 24 h for the hydrated biocrusts (yellow). Gray shading indicates nighttime incubation conditions.

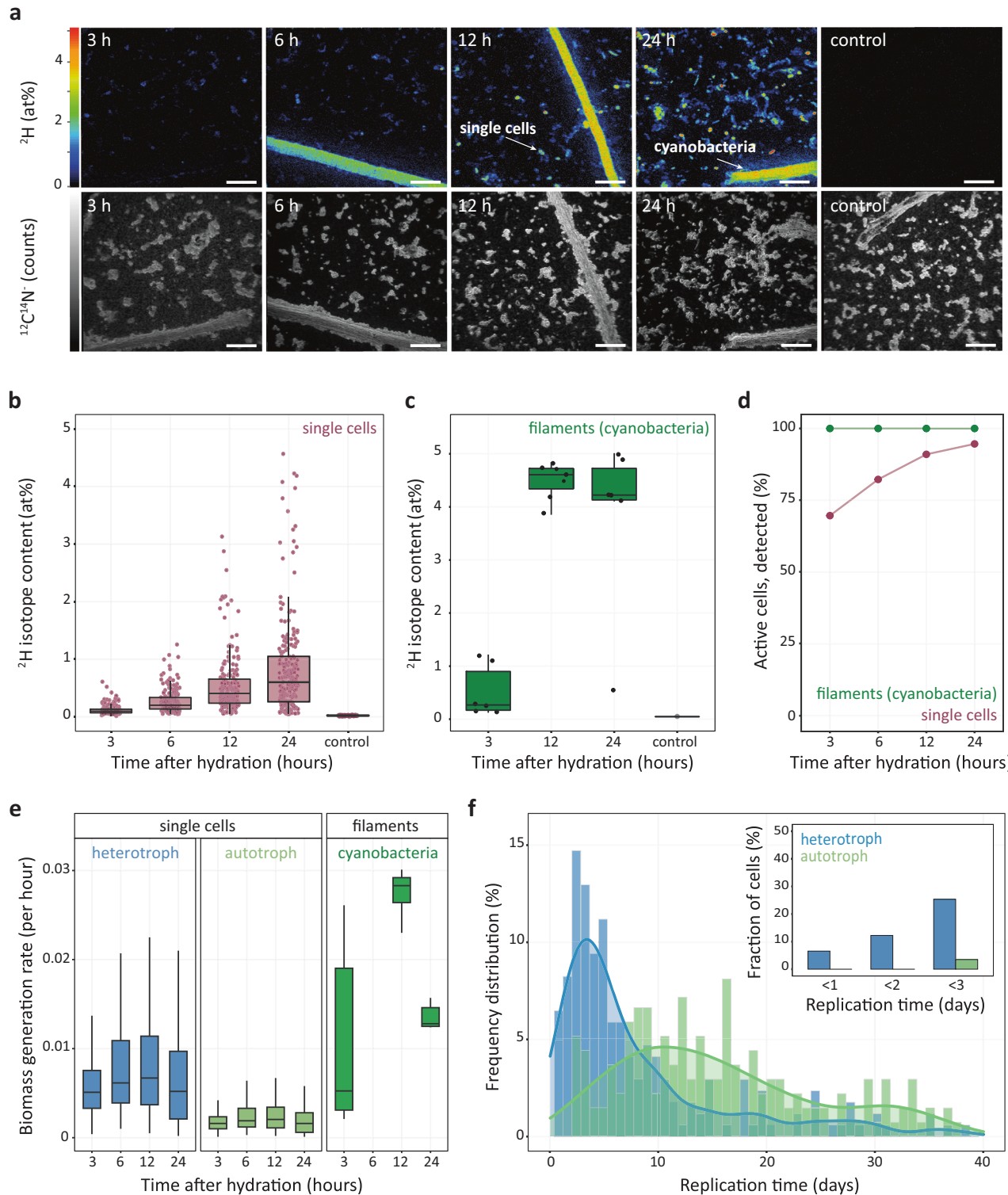

cyanobacteria after 12 h of incubation (Fig. 2a–c and Supplementary Data 1).

Based on these $^2$H enrichment data that stem from multiple time points during the 24 h period of hydrated crusts, we calculated biomass generation rates, which varied over time in filamentous cyanobacteria (Fig. 2e). In single cells, anabolic activity only showed a modest variation (Fig. 2e). Please note that sample preparation influences the isotopic composition of cells[35–38], therefore, a correction factor was applied to account for a dilution in $^2$H (see "Methods", Supplementary Note 1, Supplementary Fig. 2 for details). To cover the

metabolic diversity found in biocrust single cells (including autotrophs, mixotrophs, and heterotrophs[20]) and considering the untargeted NanoSIMS approach, biomass generation rates of single cells were calculated for either a chemoautotrophic or heterotrophic metabolism[36,39] (see "Methods" and Supplementary Note 1 for details). By estimating biomass generation rates considering chemoautotrophic and heterotrophic metabolisms, in which H atoms will stem either from water or mainly from organic compounds, the resulting range also encompasses rates for mixotrophs, which are particularly abundant in these biocrusts as shown in our previous metagenomic

**Fig. 2 | Microbial activity detected through cellular incorporation of $^2$H and NanoSIMS analysis. a** NanoSIMS images showing the $^2$H isotope content and $^{12}C^{14}N^-$ secondary ion signal intensity distribution of samples obtained from heavy water ($^2$H$_2$O) incubations after different incubation times. **b, c** $^2$H isotope content extracted from defined regions of interest (ROIs) of single cells and multicellular cyanobacterial filaments after sampling at four (single cells) or three (filaments) time points. Number of displayed cells or cyanobacterial filaments are the following: $n_{single\_cells}$ (3 h) = 164, $n_{single\_cells}$ (6 h) = 204, $n_{single\_cells}$ (12 h) = 192, $n_{single\_cells}$ (24 h) = 229, $n_{single\_cells}$ (24 h control) = 178, $n_{filaments}$ (3 h) = 6, $n_{filaments}$ (12 h) = 7, $n_{filaments}$ (24 h) = 6, $n_{filaments}$ (24 h control) = 1. **d** Fraction of cells classified as anabolically active. **e** Calculated biomass generation rates, inferred from NanoSIMS measurement data of single cells after different incubation times (and classified as

active, shown in (**b**) and (**c**)), assuming either a heterotrophic (left panel) or chemoautotrophic (central panel) physiology and of photoautotrophic cyanobacterial filaments (right panel). Outliers are not displayed. **f** Histogram visualizing the frequency of replication times of single cells based on the assumption that all cells exhibit either a heterotrophic or chemoautotrophic physiology. Smoothened lines indicate kernel density estimates. Displayed data are based on the 24 h incubation sample and cover 91% (assumed heterotrophic physiology) and 72% (assumed chemoautotrophic physiology) of cells exhibiting replication times up to 40 days. The inset depicts the fractions of cells that potentially replicate in 1, 2, and 3 days. Scale bars in (**a**) correspond to 5 μm. The boxes in (**b, c, e**) comprise the 2nd and 3rd quartiles with the horizontal line indicating the median. Whiskers maximally extend to 1.5 times the inter-quartile range.

investigation[20]. Converting these rates of single cells into cellular replication times, they ranged from hours to hundreds of days (Fig. 2f and Supplementary Fig. 3). More specifically, calculating replication times for a heterotrophic metabolism yielded a median doubling time of 5.6 days, ranging from as short as 7 h to as long as 147 days. When assuming a chemoautotrophic metabolism, the median doubling time increased to 18.7 days, with a minimum of 2.1 days and a maximum of 471 days. Replication times of filamentous cyanobacteria ranged from 0.96 to 21 days.

## Rapid reactivation of diverse microbial populations upon rehydration

Genome-resolved metatranscriptomics revealed rapid and simultaneous reactivation of diverse biocrust microbial community members upon rehydration, with distinct transcription patterns depending on hydration phase. We sequenced metatranscriptomes from a rehydration experiment, in which biocrusts were exposed to a simulated rain event that hydrated the samples for 24 h, followed by a desiccation phase. Samples were sequenced at 15 and 30 min, and 3, 6 and 12 h hydration times, and at time points 39 and 55 h that included a desiccation period of 15 and 31 h, respectively, after the 24 h hydration period (for more details, see Fig. 1 and Supplementary Table 2). In order to resolve changes in transcription in individual populations, we mapped the transcriptomic reads to previously generated metagenome-assembled genomes (MAGs) (representing all major microbial taxa in the biocrust community[20], Supplementary Fig. 4) from the same sample material and rehydration experiment. The following results refer to transcriptomic responses to hydration in microbial populations represented by the 96 MAGs. For an overview of the bulk data and its relation to the MAGs see Supplementary Note 2. Among the MAGs, two phyla recruited the most transcripts, namely *Cyanobacteria* (13–76%, with Microcoleus01 MAG recruiting 11–72%) and *Actinobacteriota* (7–43%, with Rubrobacter01 MAG recruiting 3–17%) (Supplementary Data 2). Among the analyzed hydration time points, no systematic taxonomic shifts in transcriptional activity were detectable (Fig. 3a, left panel).

Samples grouped by time since hydration when clustered based on relative abundances of individual gene transcripts (Fig. 3a, middle panel), and this trend was even more evident when relative transcript abundances were normalized for each MAG's share of the overall transcriptome (Fig. 3a, right panel). Three distinct sample clusters were identified corresponding to three specific time periods and hydration states of the experiment (ANOSIM *R*: 0.96, *p* value = 0.0001, Fig. 3a, right panel): (1) dry phase (water content below 6% at 0, 39 and 55 h); (2) early hydration phase (water content ~25% between 15 and 30 min); and (3) main hydration phase (water content ranging from 23.5% ± 0.5% at 3 h to 18% ± 2% at 12 h, Supplementary Table 2). Clustering could not separate the 39- and 55-h time points, or time points 3, 6, and 12 h after hydration (Supplementary Fig. 5), suggesting that there were no systematic changes in transcription among these time points. It is worth noting that within the cluster of dry samples, the 0-h

samples clustered separately from the 39- and 55-h samples, indicating a slightly different expression profile before and after the experiment (for discussion see Supplementary Note 2).

Normalization of transcripts per MAG allowed us to investigate differential expression of genes between the different time points and phases of the experiment within individual microbial populations and revealed rapid and collective reactivation of microbial populations upon rehydration, independent of taxonomy or physiology. The most significant changes in transcription occurred between 0 and 15 min, 30 min and 3 h, and 12 and 39 h (Fig. 3b), delineating the above-described phases of the experiment. More specifically, 85 out of 96 populations exhibited significant changes in relative transcript abundance of at least some genes (DeSeq2 adj. *p* value < 0.05) within the first 15 min of hydration (Fig. 3c, inner shaded circle), and 94 out of 96 populations among any subsequent sampling time points or among different phases of the time series (Supplementary Data 2). Only one *Gamma*- and one *Alphaproteobacteria* population had no significant differentially expressed genes during the entire time series. Subsequently, we explored transcripts indicative of specific metabolisms in individual MAGs and significant changes (DeSeq2 adj. *p* value < 0.05) in their expression between time points or phases (only observations that refer to significant differentially expressed genes are mentioned below).

## Microbial populations generate energy and repair DNA in the onset of resuscitation

In the early hydration phase (15 to 30 min after the rain event), transcripts for DNA repair and energy generation were significantly increased among microbial populations. Transcripts involved in repair of double-stranded DNA breaks, a common and challenging type of DNA damage during desiccation[40], were more abundant across numerous taxonomic groups (10 *Rubrobacteria*, 9 *Chloroflexi*, 2 *Cyanobacteria*, 2 *Bacteroidota*, 2 *Alphaproteobacteria*, a *Gemmatimonadota*, a *Deinococcota* and a *Verrucomicrobia* MAG) (Fig. 4a and Supplementary Data 3). In several populations, this coincided with the higher relative expression of terminal oxidases used in aerobic respiration (such as cytochrome c and cytochrome bd), namely, in 7 *Rubrobacteria*, 6 *Alphaproteobacteria*, a *Gemmatimonadota*, a *Bacteroidota*, and a *Cyanobacteria* population (Fig. 4b). In other microbial populations, genes involved in DNA repair and energy generation received higher transcript proportions in the main hydration phase (Fig. 4a, b).

Energy for DNA repair could stem from the degradation of storage compounds like polyhydroxy-alkanoates (PHA), and accordingly, PHA-metabolism related transcripts significantly increased in abundance in the early hydration phase in 6 *Rubrobacteria* and 4 *Alphaproteobacteria* MAGs (Fig. 4c). Transcripts involved in degradation of glycogen, another common storage compound, rather reached higher proportions in the main hydration phase or dry phase (Fig. 4c). Transcripts encoding organic compound transporters revealed highest transcript abundance in the main hydration phase (Supplementary Fig. 6 and Supplementary Data 3), suggesting that for many

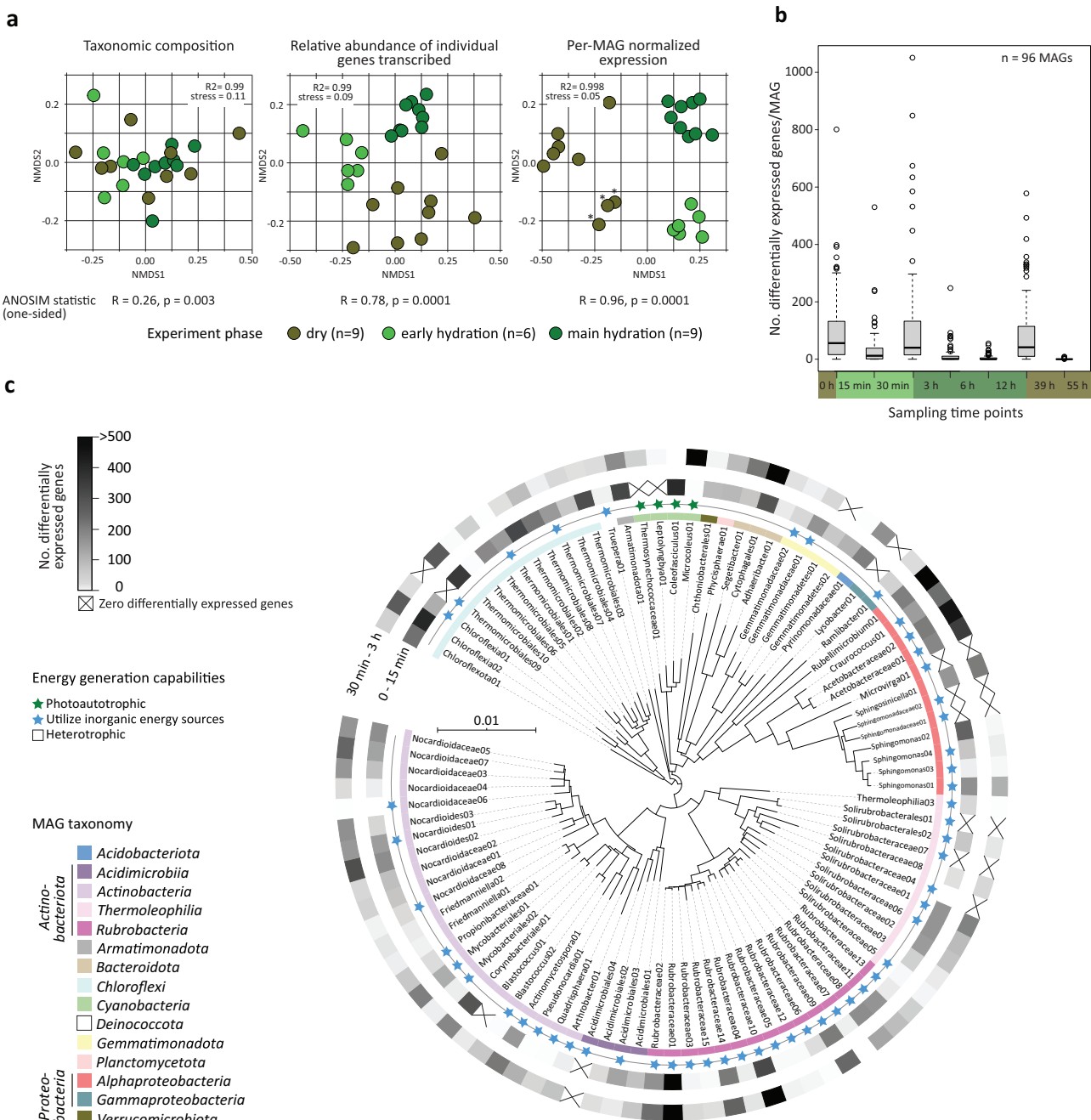

**Fig. 3 | Temporal changes in metatranscriptome composition during hydration and dehydration of biocrusts. a** Ordination of metatranscriptomes by non-linear multi-dimensional scaling (NMDS, Jaccard distances) based on taxonomic composition (left panel), relative abundance of individual genes transcribed (middle panel), and per-MAG normalized expression (right panel). Asterisks mark samples of time point 0 (dry conditions at beginning of the experiment). A one-sided analysis of similarities (ANOSIM) was performed on sample groups as indicated below the panel. **b** Number of significantly differentially expressed genes per MAG between time points. The boxplots summarize data from 96 MAGs at each transition. The boxes comprise the 2nd and 3rd quartiles with the horizontal line indicating the median. Whiskers maximally extend to 1.5 times the inter-quartile range or to the last value within that range. Note that most changes in transcription occur when transitioning between hydration phases as seen in (**a**). **c** Number of significantly differentially expressed genes (DeSeq2 adj. $p < 0.05$) per individual MAG comparing early time points after rehydration ($n = 3$ independent crust samples per time point). Effect sizes (as Log2-fold change) and Benjamini–Hochberg false discovery rate adjusted $p$ values (calculated with DeSeq2) for analyzed genes indicative of discussed metabolisms, can be found in Supplementary Data 3. The phylogenetic tree, based on GTDB-Tk[106] placement of the MAGs, illustrates the diversity of MAGs for which transcription of genes was examined. Note that biocrust populations show a transcriptional reaction irrespective of being photoautotrophic (marked by green star), mixotrophic (marked by blue star) or purely heterotrophic (no star).

populations, external organic energy sources do not play a key role in early resuscitation metabolism. Bacteriorhodopsin genes received significantly higher transcript proportions in the early hydration phase in four MAGs (all *Rubrobacteria*) (Fig. 4c), suggesting that light could be an additional energy source at this stage.

## Microbial populations acquire carbon and energy during the main hydration phase

During the main hydration phase (3, 6 and 12 h after the rain event), where most of the microbial community is actively generating new biomass (Fig. 2), the populations steadily expressed genes for energy

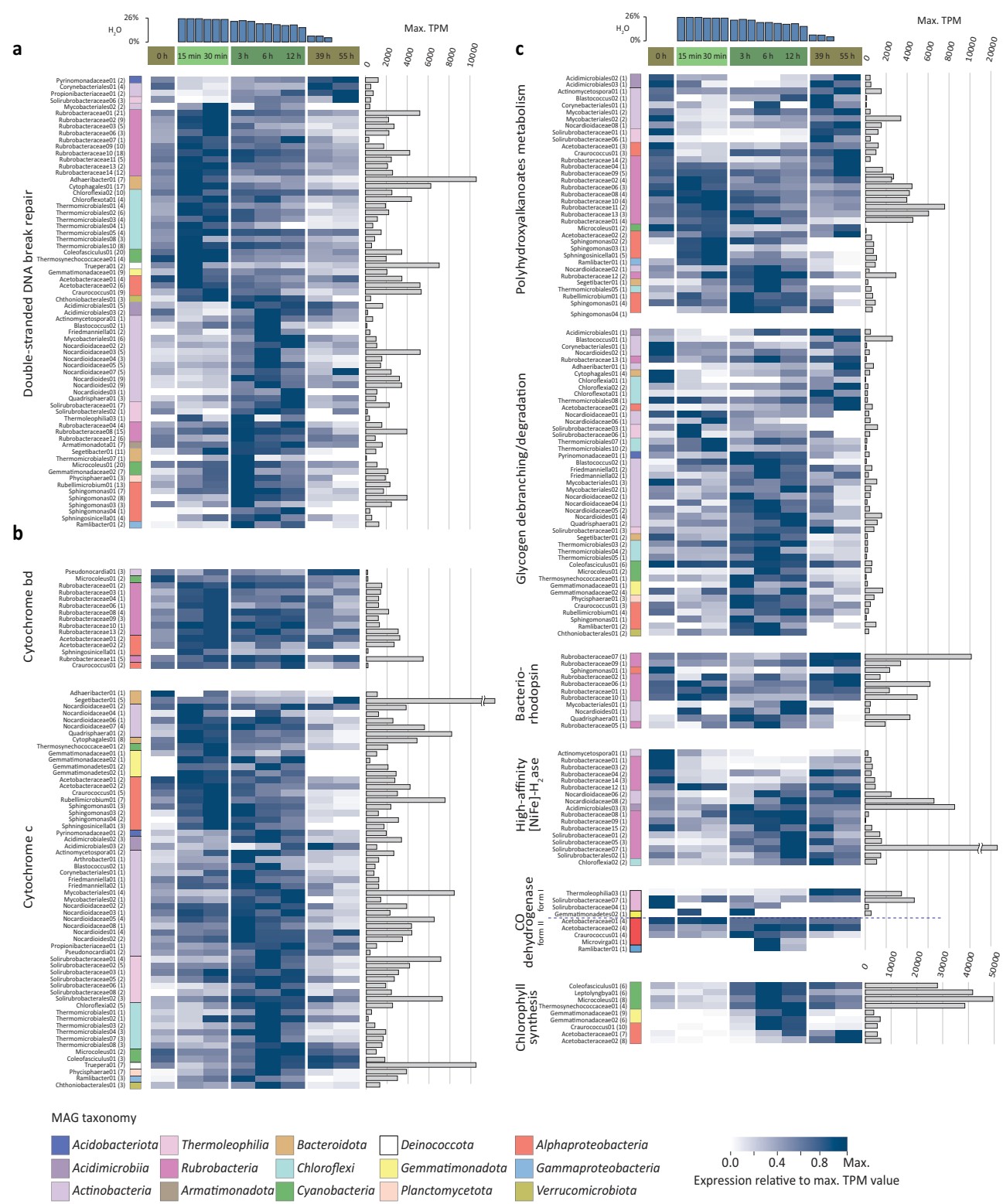

generation (Fig. 4b) and carbon acquisition (Supplementary Fig. 6b, c). If not already at high transcript abundance in the early hydration phase, the proportion of aerobic terminal oxidases (cytochrome c and cytochrome bd) transcripts increased significantly during the main hydration phase in multiple MAGs (41 out of 96, Fig. 4b). High respiration rates after hydration were confirmed by measuring $CO_2$ release (3720 ng C/g/h versus 0.61 ng C/g/h in the dry state). For heterotrophic microorganisms, the respired substrates could be sugars,

amino acids, peptides and polysaccharides given the increased proportion of transcripts encoding the respective transporters (Supplementary Fig. 6a, b). In mixotrophic MAGs (defining mixotrophy as the potential to use organic and/or inorganic energy and carbon sources), energy in the main hydration phase could be gained by $H_2$ oxidation or phototrophically (highest proportion of chlorophyll synthesis gene transcripts found in eight MAGs) (Fig. 4c). $H_2$-oxidation assays confirmed rapid $H_2$ oxidation in the hydrated state (Fig. 5a), and

**Fig. 4 | Relative transcript abundances of genes encoding for DNA repair and energy production across the temporal hydration phases for the phylogenetically diverse MAGs. a** Relative transcript abundances of genes involved in double-stranded DNA break repair, (**b**) genes encoding subunits of cytochrome bd and cytochrome c terminal oxidases indicative of aerobic respiration, (**c**) genes indicative of storage compounds degradation, atmospheric gas oxidation and light-dependent electron donor reactions. Only genes with a significant change in expression (DeSeq2 (Wald test) adj. $p < 0.05$) between subsequent time points ($n = 3$ independent crust samples per time point) or between experiment phases ($n = 6$ independent crust samples in early hydration phase, $n = 9$ for dry and main hydration phase) are shown, with the exception of Form I CO-dehydrogenase genes in (**c**), where expression patterns of all Form I CO-dehydrogenase genes are shown.

Effect sizes (as Log2-fold change) and Benjamini–Hochberg false discovery rate adjusted $p$ values (calculated with DeSeq2) for analyzed genes indicative of discussed metabolisms, can be found in Supplementary Data 3. Heat map columns depict individual time points of the time series (average values of three replicates), whereas rows depict transcripts attributed to a specific MAG. Taxonomy of individual MAGs are color-coded. Numbers parenthetically indicate the number of genes summarized per MAG (encoding different subunits or multiple copies of the same gene). The highest color intensity indicates the time point where the respective transcript reached its highest proportion in a MAG's transcriptome. This maximum value is indicated on the right in transcripts per million (TPM) (gray bars).

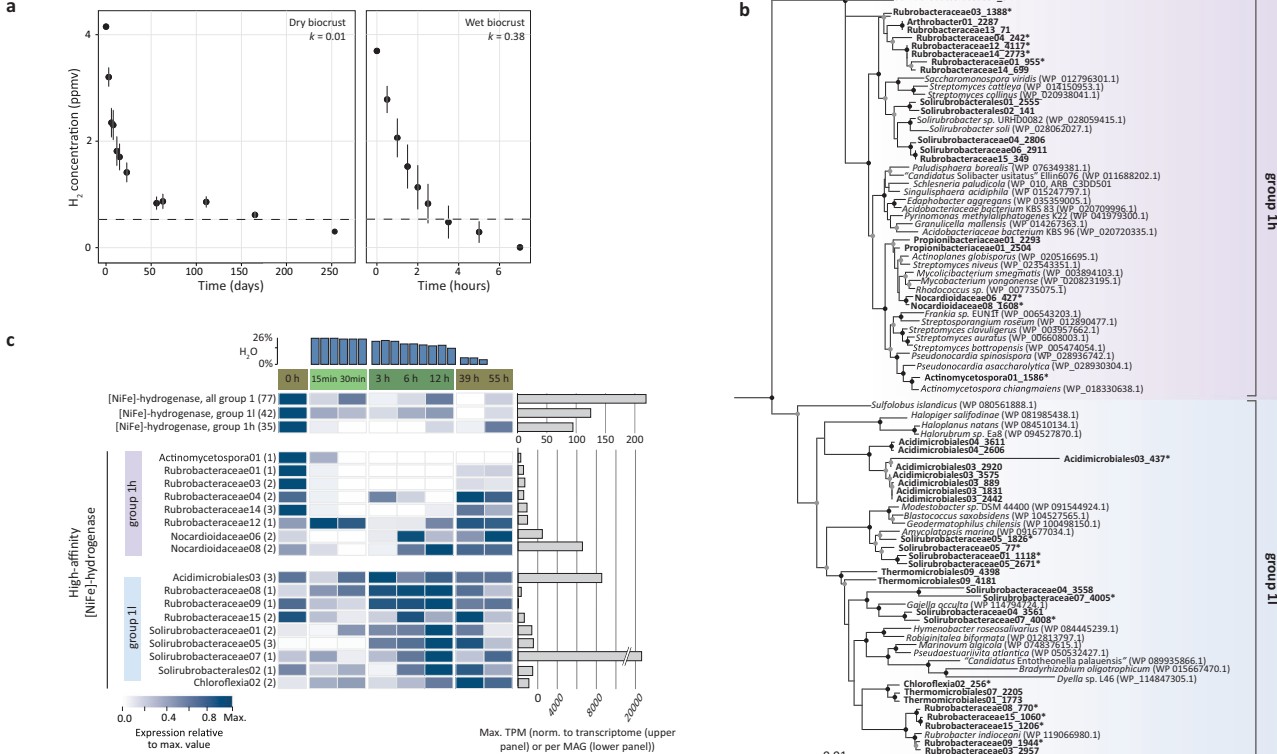

**Fig. 5 | Atmospheric H₂ oxidation, [NiFe]-hydrogenase phylogeny and gene expression in biocrusts. a** H₂ consumption over time by wet and dry biocrusts. Dotted lines represent atmospheric H₂ concentration (0.53 ppmv). Data points depict mean ± standard error of triplicates. **b** Maximum likelihood phylogenetic tree of amino acid sequences of the group 1h and 1l [NiFe]-hydrogenase large subunit (HhyL) from metagenome-assembled genomes and reference sequences. Sequences with an asterisk are MAGs with associated expression data in (**c**). Tree was generated by iQTree. The tree was bootstrapped 1000 times based on UFBoot[107], and nodes with consensus support >100% (black circle) and >85% (gray circle) are shown. Shaded sections depict the respective [NiFe]-hydrogenase

groups: purple and blue for group 1h and 1l [NiFe]-hydrogenases, respectively. [NiFe]-hydrogenase sequences from this study are in bold. Sequences from group 1g [NiFe]-hydrogenases (WP_011761956.1, WP_012349775.1) were used as an outgroup. The scale bar indicates the numbers of substitutions per site. **c** [NiFe]-hydrogenase expression patterns of the entire metatranscriptome (upper panel) and individual MAGs (lower panel) across the temporal hydration phases (average values of three replicates). Numbers parenthetically indicate the number of genes per MAG (encoding different subunits or multiple copies of the same gene). Maximum transcripts per million (TPM) are depicted.

phylogenetic analysis of the [NiFe]-hydrogenases large subunit (HhyL) showed that genes with elevated transcript proportions in the main hydration phase (in MAGs belonging to *Acidomicrobiales* and *Solirubrobacteraceae*) encode the recently described group 1l [NiFe]-hydrogenase[41,42] (Fig. 5b, c). Further, RuBisCO transcripts, indicative of autotrophic carbon fixation, showed higher proportions in the main hydration phase in eight populations (Supplementary Fig. 6c). Atmospheric carbon monoxide (CO) could be another inorganic energy source. Nine MAGs encoded either a form I or II CO-dehydrogenase (Fig. 4c), with the associated transcripts being detected throughout the experiment, mostly with significantly higher proportions of transcripts in the main hydration phase. In agreement with increased energy generation and carbon acquisition, several MAGs exhibited

higher proportions of transcripts for complex and energy demanding processes such as exopolysaccharide synthesis and motility in either the early hydration or main hydration phase (Supplementary Fig. 7 and Supplementary Data 3).

## Microbial populations are protected by accumulated antioxidants and acquire energy from inorganic sources in the dry phase

Transcripts involved in protection against reactive oxygen species (ROS) and in energy generation from inorganic sources showed distinct patterns and were particularly abundant in the dry phase (0, 39 and 55 h). We found higher proportions of manganese-based catalase transcripts in the dry phase for 37 MAGs, while only one *Rubrobacteria*

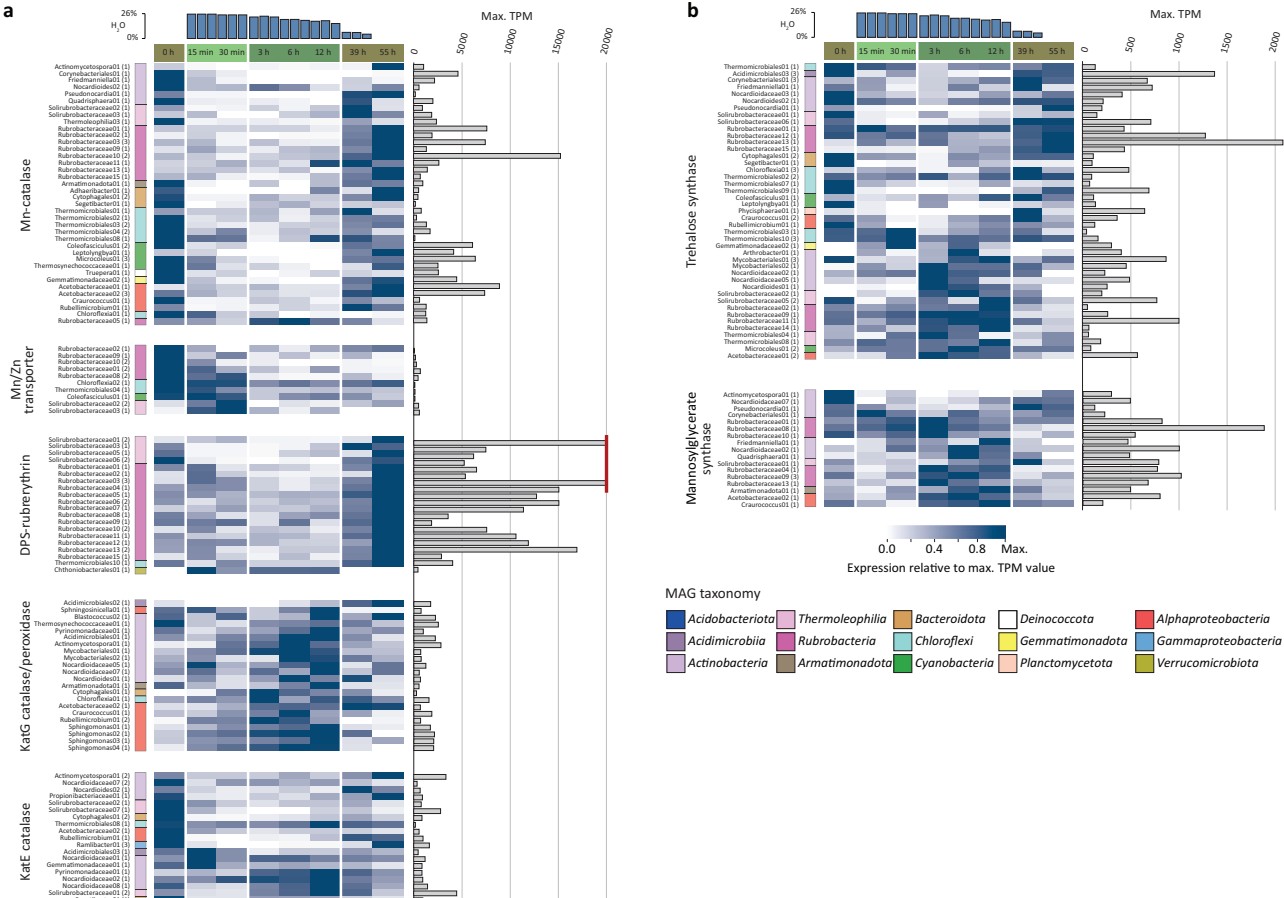

**Fig. 6 | Relative transcript abundances of genes encoding desiccation stress resistance mechanisms across the temporal hydration phases for the phylogenetically diverse MAGs. a** Relative transcript abundances of genes encoding for reactive oxygen scavenging and (**b**) osmoprotectant synthesis genes. Only genes with a significant change in expression (DeSeq2 adj. $p < 0.05$) between subsequent time points ($n = 3$ independent crust samples per time point) or between experiment phases ($n = 6$ independent crust samples in early hydration phase, $n = 9$ for dry and main hydration phase) are shown. Effect sizes (as Log2-fold change) and Benjamini–Hochberg false discovery rate adjusted $p$ values (calculated with DeSeq2) for analyzed genes indicative of discussed metabolisms, can be found in Supplementary Data 3. Heat map columns depict individual time points of the time series (average values of three replicates), whereas rows depict transcripts attributed to a specific MAG. Taxonomy of individual MAGs are color-coded. Numbers parenthetically indicate the number of genes summarized per MAGs (encoding different subunits or multiple copies of the same gene). The highest color intensity indicates the time point where the respective transcript reached its highest proportion in a MAG's transcriptome. This maximum value is indicated on the right in transcripts per million (TPM) (gray bars). The vertical red line indicates values far exceeding 20,000 TPM.

MAG recruited higher transcript proportion in the main hydration phase (Fig. 6a). Also, the transcripts of ATP-dependent transporters for manganese, intracellular accumulation of which was shown to be highly beneficial for bacterial desiccation survival[43], showed significantly higher relative abundances in the dry or early hydration phases in 10 MAGs (Fig. 6a). Transcripts of rubrerythrin, a DNA-binding protein with a catalase function that confers DNA-protection-during-starvation (DPS)[44] showed significantly higher relative abundances in the dry phase in 19 out of 20 MAGs that harbored this gene (Fig. 6a). In contrast, the heme-based KatG catalase/peroxidase of 18 from 22 MAGs with a *katG* gene had higher transcript proportions in hydrated samples (Fig. 6a), suggesting that KatG removes ROS in the phase of high metabolic activity. Heme-based KatE catalase transcripts showed a more divergent pattern with many MAGs having higher proportions of KatE-transcripts during the dry phase while others during the hydrated phases (Fig. 6a).

When looking at synthesis of small organic osmoprotectant molecules that can stabilize macromolecules by replacing water molecules in the hydration layer[45], we found highest proportions of trehalose synthase transcripts in the dry phase in 24 MAGs and in the early or main hydration phase in 16 MAGs (Fig. 6b). Notably,

*Rubrobacteria* populations, which are also known to utilize mannosylglycerate as osmoprotectant, were mainly expressing trehalose and mannosylglycerate synthase in the main hydration phase (4 and 6 MAGs, respectively), while the most abundant MAG (Rubrobacteraceae01, based on metagenomic coverage[20]) expressed trehalose synthase throughout the hydration-desiccation cycle. Genes encoding transporters for the osmolyte glycine betaine showed highest transcript abundance in the dry phase only in 9 MAGs, while for 33 MAGs highest abundances were detected in the main hydration phase (Supplementary Fig. 8 and Supplementary Data 3).

Recently, it was suggested that energy demands during desiccation can be met by using inorganic energy sources such as light and atmospheric gasses[46]. Proportions of transcripts that encode bacteriorhodopsins, which can function as light-driven proton pumps or be involved in sensory processes[47], were significantly different between the experiment phases in 11 MAGs (Fig. 4c). Several *Rubrobacteria* MAGs and a *Sphingomonas* MAG (*Alphaproteobacteria*) showed higher proportions of bacteriorhodopsin transcripts in the dry phase or early hydration phase (Fig. 4c), while three *Actinobacteria* and one *Rubrobacteria* MAG showed higher proportions in the main hydration phase. Of 12 MAGs that encode the heliorhodopsins, a recently discovered

rhodopsin without any ion pumping activity and a yet unclear function[48], only six actively transcribed it (Supplementary Data 3) and only one *Chloroflexi* MAG showed significant changes in transcription with higher proportions of heliorhodopsin transcripts in the dry phase.

$H_2$ uptake in dry biocrusts was detected (Fig. 5a), and multiple MAGs recruited highest proportions of [NiFe]-hydrogenase transcripts in the dry phase (Figs. 4c and 5c), supporting the conjecture that this process contributes to persistence during periods of starvation[42,49–54]. Microbial activity in the dry state was confirmed by low, but detectable respiration (0.61 ng C/g/h). Phylogenetic analysis of the [NiFe]-hydrogenases large subunit (HhyL) revealed that these genes encode the group 1h [NiFe]-hydrogenase (Fig. 5b), for example of four *Rubrobacteria* MAGs. However, group 1h [NiFe]-hydrogenase genes of two *Nocardioidaceae* MAGs (*Actinobacteria*) exhibited higher transcript proportion in the main hydration phase and in 39- and 55-h dry phase samples (Fig. 5c).

By combining genome-resolved gene expression with $H_2$ oxidation data, we provide in situ evidence of specific actively $H_2$-oxidizing populations. Based on these data we propose that different enzyme groups enable $H_2$ oxidation under differing conditions, namely survival or mixotrophic growth. Group 1l [NiFe]-hydrogenases exhibited higher transcript proportions in the hydrated states, whereas group 1h [NiFe]-hydrogenases had higher expression in the dry state (Fig. 5c). Such in situ evidence on differential gene expression of both hydrogenase groups required genome-resolved metatranscriptomics, as previous bulk metatranscriptomics could not detect distinct transcription patterns in dry and hydrated desert soils[55]. Faster $H_2$ consumption was detected in the hydrated state compared to the dry state, in congruence with other arid soils[34]. Considering the observed differential rate constant ($k$) values of $H_2$ uptake in dry biocrusts ($k = 0.01$) as compared to hydrated biocrusts ($k = 0.38$) (Fig. 5a), we suggest a differential contribution of the group 1h and 1l [NiFe]-hydrogenases. We hypothesize that group 1h [NiFe]-hydrogenases are mainly utilized for energy generation during starvation, while group 1l [NiFe]-hydrogenases are also involved in mixotrophic metabolism during conditions supporting growth. We extrapolate that this mixotrophic nature of group 1l [NiFe]-hydrogenases could extend to other environments, as we observed a high similarity of group 1l sequences recovered from the Negev Desert biocrust and other previously investigated soils[41,42].

## Discussion

The combination of genome-resolved metatranscriptomics and single-cell activity analysis offered unprecedented insights into microbial resuscitation dynamics and the underlying molecular mechanisms following simulated rainfall events in a diverse arid biocrust microbial community. The microbial community was able to resuscitate rapidly (within 15 to 30 min) and collectively (>90% of the cells producing biomass). There were no taxonomic patterns regarding the reactivation speed, as rapid resuscitation was detected by metatranscriptome data in almost all taxonomically and physiologically diverse populations.

Such fast response to rehydration was previously only known for biocrust cyanobacteria[25,26], while information was lacking for the remaining majority of the diverse microbial community. Our study fills this knowledge gap and additionally reveals metabolic processes that fuel resuscitation. For example, energy generating processes (such as respiration and storage compound degradation) were stimulated just minutes after rehydration, characterizing the early hydration phase (Fig. 7). Transcripts indicative of energy generation from light (via rhodopsins) and atmospheric gases (mainly group 1h [NiFe]-hydrogenase genes) (Fig. 4c) were already present since the dry phase. This immediate access to energy facilitated processes critical to survival, i.e., DNA repair. Our experimental set-up and genome-resolved metatranscriptome analysis enabled observation of transcription patterns that were previously hidden in studies that merged transcripts across large taxonomic groups (like *Cyanobacteria* and *Actinobacteria)* and broad functional categories[56,57], or focused on the expression of specific metabolic marker genes[58]. Thereby, we revealed distinct phases of activity in a rain event previously only observed in cyanobacteria[25,26] (Fig. 7).

Our observed simultaneous resuscitation of desert biocrusts microbial community members contrast with previous work on semi-arid soils from a Californian grassland, where reactivation and growth followed taxonomic patterns[12,13]. We postulate that the contrasting patterns of reactivation are caused by different soil strata investigated, as in the Californian grassland, the top 5 to 10 cm of soil were studied rather than a biocrust. That soil strata holds intrinsically different microbial community compositions and are e.g., dominated by members of the *Acidobacteriota*[59], which represent only a minor fraction of the Negev Desert biocrust communities (Supplementary Fig. 4). Furthermore, edaphic properties differ in the two systems, which could influence hydration and thus resuscitation. Hydration of biocrusts is influenced by morphology and biomass[30], and strongly by hygroscopic extracellular polymeric substances (EPS)[60], which could result in more homogenous hydration for the biocrusts microorganisms. For instance, fast water saturation was previously reported in biocrusts further west in the Negev Desert[61,62]. In contrast, hydration in Californian grasslands topsoil could be influenced by different sized pores, likely resulting in heterogeneous rewetting[63–65]. As such, differences in hydration speed could further explain the observed resuscitation patterns. In addition, these contrasting patterns could also result from the application of different methodological approaches with different sensitivities of anabolic activity detection: growth-based (rRNA content and synthesis of DNA) approaches[12,13] versus transcriptional changes in our study that are not necessarily related to growth (such as energy generation or repair).

There was no interdependence among members of the microbial community during the resuscitation process. Heterotrophs such as carbohydrate utilizing *Bacteroidota* resuscitated as quickly as photoautotrophic cyanobacteria. Initial resuscitation (early hydration phase) was accompanied by oxidation of the readily available organic storage compound PHA and use of inorganic energy sources such as light in many populations (Fig. 4 and Fig. 7). This pattern is further supported by the majority of organic compound transporters being expressed in the main hydration phase rather than during early resuscitation (Supplementary Fig. 6a, b and Fig. 7). This apparent independence from external organic compounds for initial resuscitation (Fig. 7) matches the relatively low organic carbon content in these biocrusts (0.60% in our experiment, 0.31 to 0.86% previously reported[66]). The early hydration phase might reflect a lag phase of microbial growth, in which readily available energy sources in form of storage compounds are used for immediate resuscitation e.g., to power repair of damaged cell components, congruent with culture-based studies[67,68]. The expression of RuBisCO by several actinobacterial and alphaproteobacterial populations, indicating photo- or chemolithoautotrophic $CO_2$-fixation, suggests an even higher degree of independence from phototrophic *Cyanobacteria*. Taken together, this study challenges the perception of the previously described dependency of heterotrophs on phototrophic primary producers in biocrust[69,70]. We hypothesize that this dependency does not exist in a single resuscitation event. We rather consider it as a long-term systemic dependency, as storage compounds that enable the resuscitation of heterotrophs are ultimately built up over time, from carbon fixed by autotrophic organisms.

Remaining in a dormant state after hydration is not a common strategy in arid biocrust microorganisms, as almost all biocrust microbial cells synthesized new biomass within hours after rehydration (Fig. 7). These are the highest numbers of active cells reported for soils, as previous studies reported numbers of ~2% or at maximum

**Fig. 7 | Microbial processes in dry biocrusts and following a rain event.** Conceptual figure summarizing the observed resuscitation patterns of biocrust microorganisms, including metabolic processes during dry and hydration phases. Rewetting of biocrusts will stimulate microbial activity, driving major ecosystem processes (e.g., $H_2$ oxidation, respiration, photosynthesis). Nearly all cells will become anabolically active in a rain event, but short rain phases only allow cell division in a small proportion of cells. However, even non-growing cells can use the hydration phase for repairing macromolecules and replenish reserves (such as storage compounds), which increases the chance that they persist until the next rain event.

$25\%$[71], the latter in activated soil samples. A notable exception is a study by Couradeau et al., in which 20 to 60% (maximum) active cells were reported[72]. The minor fraction of biocrust cells that did not show activity after hydration could have been dead, extremely slow metabolizing or in a dormant state that needed specific cues for resuscitation. Based on our data, we propose that desert biocrust microbial communities are adapted to respond to infrequent rain events with rapid resuscitation to efficiently utilize the short hydration window for active metabolism. Thus, despite being limited to short windows of activity, resuscitation of nearly the complete microbial community ensures the execution of dryland ecosystem processes (Fig. 7).

The observed fast resuscitation of Negev Desert biocrust cells was coupled to slow growth with median replication times of 5.6 to 18.7 days (heterotrophic or chemoautotrophic cells), reaching estimated replication times of up to 471 days (Supplementary Fig. 3a) that are comparable to replication times recently obtained from temperate and tundra soils[73]. Based on precipitation data recorded in the central Negev Desert for the last 6 years, the vast majority of rain events lasted only 1 day (Supplementary Table 1). In such short rain events, at most 7% of the cells would be able to divide (Fig. 2f, inset). Even when considering the next most frequent rain events (2 days) or extended water retention due to EPS in biocrusts[60], only at maximum 12% of cells would be able to double in a 2-day period of wetted biocrusts. To our knowledge, these are the first growth measurements for dryland soil microorganisms in situ and do not support the expectation of major cell loss in desiccation/rehydration cycles due to sudden changes in soil water potential. Microbial biomass and diversity could not be maintained in biocrusts if in each cycle a considerable fraction of the cells dies that cannot be restored in the hydration phase. Our data reveal a system of limited microbial productivity, considering the growth estimates and the number of days in which rain allows for growth (Supplementary Table 1). Until now, information on microbial productivity in desert soil was lacking besides limited knowledge on growth of lichens in the Negev Desert[74]. Thus, our data fill this knowledge gap on dryland productivity and explain the observed slow recovery of disturbed biocrusts[75], further stressing the need for biocrust preservation.

Considering the above, biocrust microorganisms must be well adapted to the stressors they experience in deserts, namely high temperature and radiation, extended droughts, and infrequent and sudden rain events. Our metatranscriptome data support this conclusion in that they suggest various mechanisms to preserve cell structure and macromolecules in the desiccated state, without elaborate morphological transformations, enabling constant preparedness for sudden resuscitation (Fig. 7). Instead of forming resting stages (such as spores) and germinating, the strategy of many biocrust microorganisms seems to focus on protecting the cell components from oxidative damage during the dry phase by mechanisms such as accumulation of manganese ions previously observed in model organism cultures[43,76,77], manganese-based catalases, and DPS-rubrerythrin[44]. Due to this protection of proteins and DNA, the damage that still occurs during desiccation can be repaired rapidly

upon rehydration as we see in the expression patterns of DNA-repair genes. To speed up repair, the microorganisms do not rely on energy-consuming import of extracellular organic energy sources but respire readily available intracellular storage molecules like PHA or use inorganic energy sources (Fig. 7). The fact that we barely observed an upregulation of organic compound transporters (including specific osmolyte transporters, Supplementary Figs. 6 and 8) in the early hydration phase also supports the notion that there is no significant osmolyte expulsion upon rehydration[78]. Instead, osmolytes might be part of the compounds respired during this early resuscitation stage. Compared to the complex process of sporulation, such cell preservation and resuscitation strategy enables faster revival and allows one to make full use of the short hydration window as seen in our transcriptomic and NanoSIMS data. Furthermore, biocrust microorganisms seem well prepared for rapid desiccation. Transcripts of genes encoding enzymes for osmoprotectant synthesis (mannosylglycerate and trehalose) were present in dry and hydrated states in diverse populations, including *Rubrobacteria* (Fig. 6b). This suggests that the cells are maintaining a constant high level of osmoprotectant under different osmotic conditions, as was previously found in *Rubrobacter xylanophilus* cultures[79]. Based on these data we hypothesize that biocrust microorganisms do not necessarily rely on specific triggers to start preparing for desiccation but are always ready to dehydrate without taking critical damage. The observed collective resuscitation strategy maximizes the benefit of rare and short water pulses on the cellular level, as it facilitates repair of macromolecules such as DNA and replenishment of energy reserves (such as storage compounds), allowing them to persevere until the next rain event.

In drylands, soil processes depend on short hydration windows that allow for microbial activity and as such, understanding of resuscitation patterns and mechanisms is highly essential. In summary, our study reveals a desert biocrust microbial community adapted to unpredictable and short-lived rain events via immediate and simultaneous resuscitation of the majority of cells and taxonomical and physiological diverse groups. Preparation for sudden osmotic changes protects from significant cell loss and enables long-term survival. These characteristic survival mechanisms raise the question how less adapted soil microbial communities will react when faced with increased desiccation stress due to the expansion of drylands. Further, the observed limited microbial productivity supports the need for biocrust preservation and restoration, as biocrusts are essential for desert soil stabilization.

## Methods
Information on used chemicals and primers are provided in Supplementary Data 4.

### Biocrusts sample collection
Biocrust samples (~5 mm in thickness) were collected at the long-term ecological research site (LTER) Avdat (30°47′N, 34°46′E) in the central Negev Desert, Israel, during the dry seasons in 2017, 2019 and 2021. More details on the site can be found in refs. 66,80,81. In short, annual precipitation is on average 90–100 mm per year[9,66]. The soil at the sampling site was a silty loam on loess (25% sand, 55% silt and 20% clay), with reported pH values ranging from 7.9 to 8.6, organic matter content of 0.6 to 1.6%, calcium carbonate content of 33.0% and total N of 0.01%[80,82,83]. The reported organic carbon content in the biocrust was with 0.31 to 0.86% approximately twice as high as in the underlying soil[66]. All samples were stored dry at 18 °C in the dark prior to the start of the incubations.

### Microcosm incubations with heavy water ($^2H_2O$) and sample preparation for NanoSIMS imaging
A scheme of biocrust sampling during the rehydration time series can be found in Fig. 1. Approximately 2.5 g of natural dry biocrusts (sampled in September 2019) were incubated in 125 ml volume gas-tight but light-permeable microcosms and hydrated to 75% of their maximum water-holding capacity (corresponding to 26% $H_2O$/wet weight of rehydrated crust piece) by adding 30% deuterated water ($^2H_2O$) (v/v) (99.9 at%, Sigma-Aldrich, USA) dropwise to preserve the physical structure. Microcosms were incubated in a climate chamber (Aralab, Portugal), equipped with white LED lights covering the spectral range from 380 to 780 nm for 24 h under following conditions: 12 h of light (500 μmol/m²/s) at 27 °C, 1-h gradual transition, 10 h of darkness at 19 °C, and 1-h transition back to light conditions. A dead control (negative control) was prepared by incubating biocrust for 48 h in 4% (v/v) formaldehyde, followed by several washing steps in ultrapure water (MilliQ water filtered with a 0.1 μm pore size PES filter) and desiccated at 27 °C for 5 days. The control was supplemented with 30% $^2H_2O$ (v/v) and incubated for 24 h as described above. All samples (incl. the dead control) were fixed in 4% (v/v) formaldehyde for 1.5 h at room temperature, washed with ultrapure water (3x) and stored in a phosphate-buffered saline/ethanol mixture (1:1).

Time points were destructively sampled after 3, 6, 12 and 24 h of hydration; at each time point, the water content was determined, and cells were prepared for single-cell analysis by NanoSIMS. Biocrust single-cells and cyanobacterial filaments were separated from 2.5 g of biological soil crust following[84] using a modified cell detachment solution consisting of 0.35% (w/v) polyvinylpyrrolidone, 3 mM sodium pyrophosphate, and 0.5% (v/v) Tween 20 (all Sigma-Aldrich). Samples in cell detachment solution were homogenized by mixing on a stir plate for 30 min at a constant rotational speed of 300 rpm. The biocrust slurries were further homogenized by sonication using a sonication bath (Sonorex-Super_RK-31, Bandelin, Germany) at 35 kzH for 1 min to remove particles from cyanobacterial filaments. Cells were then separated from larger soil particles using Nycodenz® density gradients and centrifugation as described in ref. 84. The final concentration of Nycodenz® (Progen, Germany) was 1.42 g/ml solved in PBS. The ratio of Nycodenz and biocrust slurry was ~1:1 and subsequent centrifugation was performed with a swing-out rotor (SWT14i) on an ultracentrifuge (Beckmann Coulter, USA) at $14,000 \times g$, 4 °C and 90 min. Please note that with this approach, we are analyzing the cells that remain intact after rehydration of biocrusts and the described cell separation. Also, the above-described sample preparation can lead to a dilution of the $^2H$-content in the measured cells that is difficult to constrain. This should be considered in downstream data analysis by correcting for the dilution as described in Supplementary Note 1.

The cell fraction (supernatant and part of the Nycodenz fraction) was collected and concentrated by filtration onto gold-palladium sputter coated (film thickness ca. 120 nm) polycarbonate filters (GTTP type, 0.2 μm pore size, Millipore, USA) to a cell density of ~300 cells per 60 μm × 60 μm. Filters were extensively washed with ultrapure water to remove residual Nycodenz. Before cell staining and visualization, an array of laser marks was made on the membrane filters using a laser microdissection microscope (Leica LMD 7000, Germany). The nucleic acid stain 4,6-diamidino-2-phenylindole (DAPI, Sigma-Aldrich) was used to visualize the cells by fluorescence imaging with an inverted Leica TCS SP8X CLSM, utilizing an 100x Olympus air objective. Based on this imaging, appropriate filter regions were attached to antimony-doped silicon wafers (7.1 × 7.1 mm, Active Business Company, Germany) with a commercially available glue (SuperGlue liquid Loctide, Henkel, Germany). For more detailed imaging of cells, sections of the filters were gold coated using an Agar 108 sputter coater (Agar Scientific, Essex, UK) and images were obtained using a JEOL IT 300 scanning electron microscope at the Core Facility Cell Imaging and Ultrastructure Research (CIUS) at the University of Vienna. Approximately 240 individual single cells were targeted at each respective incubation time point and ca. 6–7 large filaments of cyanobacteria at 3, 12 and 24 h after hydration for downstream NanoSIMS analysis.

## NanoSIMS analysis

NanoSIMS measurements were carried out at the Large-Instrument Facility for Environmental and Isotope Mass Spectrometry at the University of Vienna using a NanoSIMS 50L instrument (Cameca, France). Prior to image acquisition, each analysis area was pre-conditioned with a slightly defocused $Cs^+$ ion beam to ensure minimum sample erosion during establishment of the steady state secondary ion signal intensity regime. For this purpose, the following sequence of high and extreme low $Cs^+$ ion impact energy (EXLIE) was applied: high energy (HE, 16 keV) at 100 pA beam current to a fluence of $5 \times 10^{15}$ ions per $cm^2$; EXLIE (50 eV) at 400 pA beam current to a fluence of $5 \times 10^{16}$ ions per $cm^2$; HE to an additional fluence of $2.5 \times 10^{14}$ ions per $cm^2$. Measurement strategies varied to gain either (A) high spatial resolution (low current (LC) data acquisition: 1.5 pA primary beam current, 5–7.5 ms per-pixel dwell time) or (B) high throughput (high current (HC) data acquisition: 15 pA, 1.5 ms per-pixel dwell time). Data were recorded as multilayer image stacks, each consisting of 30 to 100 individual layers. Areas between 300 and 3600 $\mu m^2$ were scanned at $256^2$ to $512^2$ pixel image resolution. The detectors of the multicollection assembly were positioned to enable parallel detection of $^1H^-$, $^2H^-$, $^{16}O^1H^-$, $^{16}O^2H^-$, $^{12}C_2^-$ and $^{12}C^{14}N^-$ secondary ions for LC measurements and $^1H^-$, $^2H^-$, $^{16}O^1H^-$, $^{16}O^2H^-$ for HC measurements (in which the $^{12}C_2^-$ and $^{12}C^{14}N^-$ secondary ions had to be excluded from detection since their signal intensities exceeded the security threshold of the electron multipliers). Secondary electrons were detected simultaneously to secondary ions to facilitate target cell identification. Measurements were conducted in triplicates on distinct analysis areas within each sample obtained from consecutive incubation times, complemented by the dead-control sample (incubated for 24 h).

Measurement data were processed using the WinImage software package provided by Cameca (version 2.0.8) and corrected for detector dead-time and image drift from layer to layer. Regions of interest (ROIs), referring to individual cells, were manually defined based on the $^{12}C^{14}N^-$ secondary ion maps and verified by the topographical/morphological appearance in the secondary electron images (Supplementary Fig. 1). For each time point (3 to 24 h after hydration of the crusts), 211–248 individual single cells and 6 to 7 cyanobacterial filaments were analyzed. Secondary ion signal intensities were corrected for quasi-simultaneous arrival (QSA) on a per-ROI basis. The QSA correction was performed applying sensitivity factors of 1.06 and 1.05 for $C_2^-$ and $CN^-$ ions, respectively. For $H^-$ ions, a factor of 1.00 was taken. The hydrogen isotope composition was inferred from the signal intensities of $^1H^-$ and $^2H^-$ secondary ions. The deuterium content, presented as the $^2H/(^1H + ^2H)$ isotopic fraction, given in at%, was determined for each ROI by averaging over the individual images of the multilayer stack. The analytical uncertainty of the measurement values, emerging from the random error in single ion counting, was estimated on basis of Poisson statistics and calculated from the $^1H^-$ and $^2H^-$ secondary ion signal intensities (given in total counts) within each individual ROI via:

$$\sigma_{\text{Poisson, at\%}\,^2H} = \frac{100}{(^1H^- + ^2H^-)^2} \sqrt{(^1H^-)^2\,^2H^- + ^1H^-\,(^2H^-)^2} \tag{1}$$

## Classification of active cells, replication time estimates and evaluation of the dynamics of anabolic activity in response to rehydration

Cells were assessed as anabolically active, if the NanoSIMS-determined mean cellular $^2H$ content exceeded the mean plus three standard deviations of the corresponding cells from the dead-control sample and if the random measurement error ($3\sigma$, Poisson Eq. (1)) was smaller than the difference between the $^2H$ content of the interrogated cell and the mean of the corresponding cells from the dead-control. The simultaneous application of these two criteria corresponds to detection of active cells at a confidence level of 99.86% (i.e.,

*p* value < 0.00135). This approach resulted in 164 single cells identified as active after 3 h, 204 after 6 h, 192 after 12 h, and 229 after 24 h of incubation (Fig. 2b).

Utilizing the NanoSIMS measurement results and following previously published approaches[36,85,86], we estimated cellular replication times and evaluated the dynamics of anabolic activity in response to rehydration. Briefly (more detailed information is provided in Supplementary Note 1), we considered the rate of cellular biomass generation $(\bar{\mu})$ as the combination of cellular growth $(\mu_{\text{growth}})$ and regeneration $(\mu_{\text{regeneration}})$ of cellular material which reads:

$$\bar{\mu} = \mu_{\text{growth}} + \mu_{\text{regeneration}} \tag{2}$$

If referred to cellular abundances, it should be noted that the number of cells remains constant for sole regeneration, whereas the number of cells exponentially increases in case of sole growth. Accordingly, the time for doubling the amount of biomass through anabolic activity is given by:

$$\tau = 1/\mu_{\text{regeneration}} \tag{3}$$

or

$$\tau = \ln(2)/\mu_{\text{growth}} \tag{4}$$

In isotope incubations coupled to isotope specific analysis, cellular growth and regeneration are indistinguishable, which means that only the combined biomass generation rate $\bar{\mu}$ (Eq. (2)) can be determined. The analysis function for inference of $\bar{\mu}$ from the measured $^2H$ content after heavy water ($^2H_2O$) incubation can be written as[73]:

$$\bar{\mu} = -\ln\left(1 - \frac{k\left(F_{\text{incub}}^{\text{NS}} - F_{\text{ctrl}}^{\text{NS}}\right)}{a_w F_{\text{D2O}} - F_{\text{H2O}}}\right)/t_{\text{incub}} \tag{5}$$

where the symbols $F_{\text{incub}}^{\text{NS}}$, $F_{\text{ctrl}}^{\text{NS}}$ refer to the isotope fraction $^2H/(^1H + ^2H)$ measured in the cells by NanoSIMS (NS) after incubation and the value determined on an inactive control sample (here, the dead control). $F_{\text{D2O}}$ and $F_{\text{H2O}}$ indicate the $^2H$ content of the isotopically enriched water used in the incubation and isotopically unlabeled water, respectively. $t_{\text{incub}}$ designates the duration of the heavy water incubation. $k$ stands for a correction factor, which considers the potential deviation of the measurement values from the actual $^2H$ content due to several factors including sample preparation (see Supplementary Note 1). $a_w$ designates the water hydrogen assimilation constant[36,86], which depends on the product of the amount of hydrogen originating from water and the isotope fractionation associated with the incorporation reaction(s). Though entering Eq. (5) as constants, $k$ and $a_w$ are subject to considerable variation. With respect to $a_w$, it is evident that the corresponding value is larger for obligate autotrophy, since the hydrogen taken up exclusively stems from water, whereas, in case of hetero- or mixotrophy, a fraction of hydrogen is also provided by organic substrate(s). Based on literature data for $k$[36] and $a_w$[39], we considered two extreme case scenarios for estimation of cellular replication times (Fig. 2e), namely obligate chemoautotrophy and obligate heterotrophy by application of $k = 1.49$, $a_w = 0.79$ and $k = 1.69$, $a_w = 0.28$, respectively. Due to lack of exact $a_w$ values for photoautotrophy, we used the values reported for chemoautotrophy for estimating biomass generation rates of phototrophic cyanobacteria. Consequently, photoautotrophic growth rates could be underestimated, as they reportedly lay within the range between chemoautotrophs and heterotrophs[39]. In all cases, the inferred biomass generation rates were exclusively ascribed to growth (Eqs. (2) and (4)). Estimation of replication times was conducted to specifically answer the question whether the cells have, based on current knowledge

about inference of metabolic rates from heavy water incubations, the capacity to double within the time period of a typical rain event.

## Hydration time series and selection of samples for RNA sequencing

A hydration experiment with $H_2O$ was established to explore the gene expression during a hydration-desiccation event (scheme depicting sampling time points during the experiment can be found in Fig. 1). Biological replicates of biocrust samples (sampled in June 2017) were chosen based on similarity of microbial community composition (Supplementary Fig. 4). Briefly, intact dry biocrusts (1.5 to 2.8 g dry weight) were rehydrated to 75% of their water holding capacity (26% wet weight) and placed in petri-dishes sealed with parafilm to avoid rapid desiccation. Crusts were incubated in a climate chamber as described above. After 24 h, the parafilm seal was removed and crusts slowly desiccated for additional 31 h. Samples were taken 15 and 30 min 3, 6, 12, 39 and 55 h after hydration in five replicates, with samples at 39 and 55 h stemming from the desiccation phase. At each sampling event, crusts were weighed to determine the remaining water content, placed into sterile 15 ml tubes, shock-frozen in liquid nitrogen and stored at −80 °C until nucleic acid extractions. For sequencing, triplicates were selected per time point based on the most similar water content (Supplementary Table 2).

## RNA extraction and sequencing

For each replicate, RNA was extracted from ca. 2.5 g of biocrust using a modified phenol-chloroform based extraction[87,88] at acidic pH with three rounds of mechanical disruption[89] via bead beating with a reduced speed of 5 m/s (FastPrep-24 bead better; MP Biomedicals, Germany). To optimize RNA yield and quality, centrifugation was prolonged to 1.5 h and an additional ethanol washing step was applied. RNA was purified using multiple rounds of enzymatic digestion with Turbo DNA-free kit (Invitrogen Life Technologies, USA) supplemented with RNAseOut (Invitrogen) and 1,4-dithioreitol (Sigma-Aldrich) (both at a final concentration of 2 mM). The frequency and incubation times for the DNAse treatment were extended to two rounds each with 60 min incubation time and an additional dose of DNAse after 30 min to ensure efficient removal of DNA from the biocrust total nucleic acid samples. RNA was precipitated in between DNase treatments with polyethylene glycol supplemented with 1 μl RNA-grade glycogen (Invitrogen Life Technologies) at 4 °C for 1 h (20,817 × g, 5430R centrifuge, Eppendorf, Austria), followed by two washing steps with ice-cold 70% RNA-grade ethanol (centrifugation at 4 °C, 20,817 × g, 40 min). To ensure all DNA was digested, a general 16S rRNA gene PCR was performed with two broadly inclusive primer sets (616V/1492R[90] and 515f_mod/806r_mod[91,92], 33 and 35 cycles, respectively). The quality and purity of the total RNA was analyzed with an Agilent 2100 Bioanalyzer using an Agilent RNA 6000 Pico chip (Agilent Technologies, United States). The purified RNA was sequenced on Illumina HiSeq 2500 in 125 bp paired-end mode upon rRNA-removal using the Illumina Epidemiology Gold Kit; library preparation was performed at the Vienna Biocenter Core Facility (https://www.viennabiocenter.org/vbcf/). The sequencing output is summarized in Supplementary Table 2.

## mRNA sequence processing

Sequence reads were trimmed using BBduk v.37.61 (BBtools v37.61; sourceforge.net/projects/bbmap/) with default parameters and error-corrected using Bayes-Hammer module of SPAdes assembler v.3.11[93]. All reads resembling ribosomal RNA were removed from the data set by mapping the reads to the SILVA SSU132 and LSU132[94] and the 5S rRNA database[95] with a sequences identity of >70% using BBmap v.37.61 (sourceforge.net/projects/bbmap/). The remaining reads were mapped to previously published metagenome contigs[20] with an expected identity of 99% and post-mapping cut-off of 97%. In addition, only pairs where both reads map in the correct orientation and correct insert size were accepted. The numbers of error-corrected and mapped reads are summarized in Supplementary Table 2.

In order to assign mapped reads to gene calls, the BAM files generated by BBmap were processed with featureCounts from the Subread package v. 2.0.0[96] resulting in a read count per gene per sample table, which was used for downstream processing in R v. 3.6.1. The read counts were translated into transcript counts via dividing the count values by gene length (in kbps). The transcript counts were translated to relative abundances (transcripts per million, TPM) in two ways: (1) For bulk metatranscriptome analysis so that all transcripts in one sample sum up to one million; (2) For MAG-resolved analysis, so that all transcript attributed to a MAG in a sample sum up to one million (Supplementary Data 3). These normalized values were used for heatmap generation in python using pandas v.1.3.3[97] and seaborn v.0.11.1[98] modules.

## Note on genome-resolved metatranscriptome analysis

We demonstrate that mapping mRNA reads to MAGs and normalizing the metatranscriptomes for each MAG separately is essential to decipher transcriptional patterns in complex microbial communities. Even though only a third of the MAGs published in Meier et al.[20] (available through public DNA sequence archives under project number PRJEB36534) have a completeness above 90% (83% on average, 55–100%)[20], they provide a solid backbone for normalizing differences in abundances of transcripts from different populations. While the crust material was carefully selected to represent the same type of crust and contain the same microbial populations within the rehydration experiment (Supplementary Fig. 4), the ratios between abundances of individual population genomes can vary significantly between crust replicates. Consequently, also the number of transcripts assigned to a given species differs substantially. For example, in samples of the 30 min timepoint depending on replicate, the Microcoleus01 MAG receives four times as many, three times as many, or an equal number of transcripts as the Coleofasciculus01 MAG. In the 3 h timepoint the differences are 4-fold, 10-fold, and 15-fold, depending on the replicate (see Supplementary Data 3). The ratios between transcripts of other populations vary in a similar fashion. Together with not entirely synchronized transcription changes, these fluctuations in abundance nearly completely obscure the transcriptional patterns of the community as it is evident from the NMDS ordination based on bulk transcriptomes (Fig. 3a). By mapping reads to MAGs we might miss the genes that these MAGs lack due to incompleteness. However, for the genes present in the MAGs, we can reveal the transcriptional patterns by alleviating the effect of fluctuating manifold differences in abundance of different populations. The effect can be clearly seen in MAG-normalized samples clustering by experimental phase in Fig. 3a. It shows that even MAGs obtained with relatively modest sequencing effort of five metagenomes can significantly improve the quality of metatranscriptome analysis.

## Differential expression analysis

All differential expression analyses were performed with DeSeq2 v. 1.26.0 with default settings[99], separately for each MAG, utilizing all non-normalized read counts assigned to the MAG, as DeSeq2 has internal algorithms for normalization and determines significance considering the sampling depth (number of reads mapped). Pairwise DeSeq2 comparisons were performed between subsequent time points and larger phases of the experiment as defined in Fig. 3a. All $p$ values and log2-fold changes of gene expression as calculated by DeSeq2 for the analyzed genes can be found in Supplementary Data 2. Genes referred to as "differentially expressed", "upregulated", "showing higher transcript proportions" are the ones that were identified as significantly differentially expressed between the said conditions by DeSeq2 with an adjusted $p$ value < 0.05. Any observations mentioned in the

manuscript refer to significantly differentially expressed genes as determined by DeSeq2. Unless explicitly stated otherwise, the heatmap plots only depict genes that had at least one significant expression change throughout the time series, either between subsequent time points or between phases of the experiment.

## Statistical analysis

Statistical analysis was performed in R using functions of the vegan package v. 2.5.4 (http://cran.r-project.org/package=vegan). The Jaccard distances were used to generate non-linear multi-dimensional scaling (NMDS) ordination plots with metaMDS (using monoMDS engine) and hierarchical clustering dendrograms of samples with hclust (average linkage method). The Jaccard dissimilarity was chosen as distance metric for two reasons: (1) metatranscriptomic data represent relative abundance data where many genes might not recruit any RNA reads leading to a sparse matrix with many zeros, and (2) the absence of transcripts in a sample should not receive weight when calculating similarities, since it might be simply due to insufficient sequencing depth. In order to determine if the separation of samples into groups was robust, analysis of similarities (ANOSIM) was performed using Jaccard distances with 9999 permutations.

## Biocrusts respiration measurements

Respiration was measured on dry biocrusts (sampled in April 2021, ~1% water content) and hydrated biocrusts (brought to 26% water content, Fig. 1) in triplicates by differentiating the naturally occurring $^{13}C$ isotope of carbon dioxide[100] in soil due to the copious amounts of carbonates in biocrusts. One g of biocrust was incubated (in triplicates) either hydrated or dry under day/night conditions as described above in 50 ml serum vials with synthetic air in the headspace containing 80% $N_2$ gas and 20% $O_2$ (Air Liquide, Austria). Gas samples were taken 24 h after wetting and at 4 time points during 137 days for the dry biocrust incubations. The gas samples were measured with a Gasbench II coupled to a Delta V Advantage IRMS (Thermo Fisher, Germany) and calculations were conducted following[100] to account for the contribution of inorganic C to $CO_2$ release ($f$SIC):

$$f\text{SIC} = (\delta^{13}C_{CO2} - \delta^{13}C_{SOC})/(\delta^{13}C_{SIC} - \delta^{13}C_{SOC}) \qquad (6)$$

In Eq. (6), $\delta^{13}C_{CO2}$ is the $\delta^{13}C$ value of the emitted $CO_2$, $\delta^{13}C_{SOC}$ is the $\delta^{13}C$ of soil organic carbon and $\delta^{13}C_{SIC}$ is the $\delta^{13}C$ of the inorganic C in the soil samples. To determine SOC, SIC as well as $\delta^{13}C_{SOC}$, ~150 mg of biocrusts (in triplicates) were acidified with 200 µl 6 M HCl, mixed and dried for 2 days at 60 °C. This procedure was repeated three times followed by a final drying step for 3 days. The HCl treated biocrusts as well as non-treated controls were milled and measured via IRMS.

## Biocrusts $H_2$ consumption assays

Five grams of dry biocrusts (sampled in April 2021) (~1% water content) and 1.5 grams of dry biocrust subsequently brought to ~26% water content were incubated in triplicates in sealed 110 ml serum bottles with butyl rubbers stoppers (see Supplementary Fig. 9, Supplementary Table 3 and Supplementary Note 3 for stopper comparison), flushed with synthetic air (Air Liquide) and supplemented with 4 ppmv $H_2$ (Linde, Austria). $H_2$ consumption was monitored using a gas chromatograph with a pulsed discharge helium ionization detector (model TGA-6442-W-4T-PT-2He, Valco Instruments Company Inc., USA), along with $H_2$ standards. Our calculated level of detection for $H_2$ on this gas chromatograph was 0.04 ppmv, with a level of quantitation of 0.12 ppmv. Autoclaved controls and stopper controls were run in parallel. Briefly, biocrusts were autoclaved at 121 °C, 15 lbs of pressure for 1 h. Water was added to the biocrusts after autoclaving to stimulate potential growth and then incubated in ambient light at 23 °C for ca. 24 h. The biocrusts were then re-autoclaved using the aforementioned conditions and then allowed to dry (completely) at 60 °C for 48 h.

## Phylogenetic analysis of high-affinity [NiFe]-hydrogenase large subunit sequences

Sequences encoding the [NiFe]-hydrogenase large subunit (HhyL) from[41] were retrieved as follows. The DNA sequences of the MAGs were downloaded from https://doi.org/10.6084/m9.figshare.12818810.v1 and genes were predicted de novo with Prodigal v. 2.6.3[101]. Large subunits of the [NiFe]-hydrogenase were identified among the translated amino acid sequences via an HMM search for the Pfam motif PF00374 with hmmer v. 3.2.1[102]. Deduced amino acid HhyL sequences encoded in the MAGs were aligned together with reference sequences belonging to 1h and 1l groups of [NiFe]-hydrogenase as described in ref. 42. Since the HydDB[103] cannot differentiate between group 1l and 1h [NiFe]-hydrogenases, these [NiFe]-hydrogenases were classified based on the phylogenetic groupings described in refs. 41,42. Phylogenetic trees were constructed with sequences of the large subunit of the group 1h,1l [NiFe]-hydrogenase extracted from taxonomically diverse MAGs. Two algorithms were used to ascertain the phylogenetic placement of these extracted sequences: (1) the Evolutionary Placement Algorithm (EPA) implementation in RAxML[104] as described in ref. 42 and (2) de novo maximum likelihood-based analysis along with reference sequences[41,42] in iQTree[105]. The sequences analyzed in this study were classified to either 1h or 1l group based on their consistent placement in the tree.

## Reporting summary

Further information on research design is available in the Nature Portfolio Reporting Summary linked to this article.

## Data availability

The quality-filtered mRNA reads have been uploaded to the European Nucleotide Archive under the project number PRJEB52014. The nanoSIMS data generated in this study are provided in Supplementary Data 1. Data used to generate the figures in this manuscript are summarized in the Source Data File_Figures. The previously generated metagenome assembly and metagenome-assembled genomes (Meier et al.[20]) that were used as a reference for transcript mapping are available through public DNA sequence archives under project number PRJEB36534. The following rRNA databases were used for filtering out the remaining rRNA reads from the transcriptomes: SILVA SSU132 (https://www.arb-silva.de/fileadmin/silva_databases/release_132/Exports/SILVA_132_SSURef_Nr99_tax_silva.fasta.gz), SILVA LSU132 (https://www.arb-silva.de/fileadmin/silva_databases/release_132/Exports/SILVA_132_LSURef_tax_silva.fasta.gz), 5S rRNA database (http://combio.pl/rrna/download/). Source data are provided with this paper.

## Code availability

Bash and R code of all sequence analysis steps can be found under the following link (https://doi.org/10.5281/zenodo.10657361).

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

## Acknowledgements

This work was funded by an ERC Starting grant (grant agreement number 636928 to D.W.) from the European Research Council (ERC) under the European Union's Horizon 2020 research and innovation program and supported by the Young Academy of the Austrian Academy of Sciences. This work was further funded in part by the Austrian Science Fund FWF (Grant-DOI: 10.55776/W1257 and 10.55776/DOC69). For open access purposes, the author has applied a CC BY public copyright license to any author accepted manuscript version arising from this submission. We thank the Division of Computational Systems Biology and University of Vienna for providing and maintaining excellent computation resources (Vienna Life Science Compute Cluster). Electron microscopy was performed at the Core Facility Cell Imaging and Ultra-structure Research, University of Vienna—member of the Vienna Life Science Instruments (VLSI). We thank Sean Bay, Capucine Baubin and Nimrod Wieler for helping hands during the crust sampling campaign, Christopher Panhölzl for assistance during the rehydration experiments for metatranscriptomics, Daniela Trojan for discussions on RNA extraction and purification, Andrew Giguere for strain isolation, the technical staff in the department for their excellent support and Nurit Agam and David Klepach for providing access to meteorological data of Sde Boker (Israel). And finally, we thank Martin Polz for his critical review of the manuscript.

## Author contributions

S.I., D.V.M. and D.W. designed research; S.I. and D.V.M. performed sampling with support from O.G.; S.I. and D.W. planned $D_2O$-incubation experiments, which were optimized and performed by S.I.; A.S. and A.L. performed NanoSIMS measurements; S.I., A.L. and A.S. analyzed NanoSIMS data; A.S. devised inference of biomass generation rates and replication times from NanoSIMS-measurement data; S.I., D.V.M. and D.W. planned metatranscriptome study; S.I. and D.V.M. optimized the incubation set up and performed metatranscriptome experiment; S.I. prepared samples for RNA sequencing; D.V.M. developed data analysis strategy, S.I. and D.V.M. analyzed and interpreted data; S.A.E. and. S.I. planned and performed $H_2$ oxidation experiment, S.A.E. analyzed the data; S.I. and J.S. planned respiration experiments and analyzed the data; A.R. contributed analytical tools; D.W. supervised the project; S.I., D.V.M. and D.W. wrote the original draft; S.A.E., S.I., D.V.M., A.S., and D.W. reviewed and edited the manuscript with contributions from all authors. All authors approved the final version of the manuscript.

## Competing interests

The authors declare no competing interests.
