## [Peer Review File · Nature Communications]

Survival and rapid resuscitation permit limited productivity in desert microbial communitiesReviewer #1 (Remarks to the Author): (see attached)

Review for “Desert microbial communities are characterized by low productivity and ensure microbial-mediated processes by rapid resuscitation” by Imminger *et al.* submitted to Nature Communications.

The manuscript presented by Imminger *et al.* documents the resuscitation of microbial community members from biological soil crust samples through the coupled use of Nano-SIMS and metatranscriptomics. The manuscript also relies on some of their previous data, notably metagenome-assembled genomes (MAGs), to consolidate their findings to show how nearly all community members activate immediately after a wetting event. The findings are interesting to a broad audience, the methods are robust, the writing clear and the figures are of high quality.

I have a few minor suggestions for the authors to clarify some of their results.

Firstly, the catalogue of MAGs used from the previous work (Meier *et al.*, 2021) comprises 96 populations. Checking the quality of these MAGs shows that just 2 can be considered high-quality genomes (primarily due to lacking ribosomal genes). In my opinion this is a shortcoming that should be mentioned or at least addressed by the authors. My concern is that the incompleteness of the genomes could obscure patterns relating to transcriptional activity. The average completeness of those genomes is ~83%, indicating that almost 1/5 genes could be missing from any population.

Moreover, many of the MAGs capture the same diversity repeatedly. There are, for example, 15 members of the Rubrobacteraceae. If we expect that their genomes, and therefore their responses to hydration to be even moderately similar, it would inflate their specific response to hydration relative to, say, Microcoleus of which there is only a single representative.

Secondly, it wasn't until rereading the manuscript that I realized that the present study does not generate new MAGs given the wording “Genome-resolved metatranscriptomics” and “Population-resolved metatranscriptomics” is a bit ambiguous. Did the authors try to assemble and bin genomes from the metatranscriptomic data? The depth of the sequencing data suggests that this may have been possible (Extended Data Table 2).

Moreover, what proportion of the community do the authors estimate is captured by the 96 MAGs? It is claimed that this dataset encompasses all major taxa. For example, cyanobacteria comprise a very large proportion of the metatranscriptomic data but only constitute four MAGs by my understanding. It seems that ~20% of the transcripts mapped to ‘unbinned’ community members based on Extended Data Fig. 3. I wonder who these microbes are and how their absence from the mapping analysis amplifies the signals reported for the genomes captured as MAGs.

I am not overly familiar with Nano-SIMS, however I would anticipate that non-filamentous cyanobacteria might exist in these Negev biocrust too. For example, one of the cyanobacterial MAGs is classified as a member of the *Thermosynechococcaceae* which are rod-shaped. Likewise, filamentous heterotrophs could be present in the biocrust samples (*Actinomyces* for example).

The estimate of doubling times is very interesting ecologically and has important repercussions for ecosystems processes. The doubling time estimates seem extremely high, or at least are not intuitive. The highest value of 471 days seems remarkably high to me, although this is an outlier. It's possible to estimate doubling time from the MAGs using codon usage bias estimates (e.g. gRodon: <https://github.com/jlw-ecoevo/gRodon2>) and I wonder how comparable these data would be with the high estimates presented here.

Line 119 - 121: p-values are reported yet the actual test performed is missing. Also, intriguingly, all p-values are identical ($p < 0.0013$) which seems at best unlikely and at worst suspicious. Please clarify these results and the test used to generate the statistics.

Line 142: The authors state that their transcriptomics reveals reactivation of diverse microbial members after wetting. However when looking at the transcriptomic data in source data files it seems that more genes are transcribed across MAGs at 0 hours than most other time points when considering the sum of all genes transcribed per time point. This indicates that microbes are not dormant during dry condition (i.e. are actively transcribing) and that the reactivation of microbes is really a reactivation of specific pathways. Is my understanding correct here?

Line 230: The grouping of samples from time points 0, 39 and 55 hours after wetting as 'dry phase' is slightly odd. Clearly the 39 and 55 hours samples had been wetted and had subsequently dried out, however it is unknown (I assume) as to how long the time point 0 had been dry. I think the authors are grouping 'recently dried biocrust' with a 'control' prior to wetting. Another reason why this grouping doesn't please me is because there are large discrepancies in the number of genes transcribed at these different times. Taking the average of the triplicate sequencing, the time 0 has ~25,000 genes transcribed across all MAGs while this value plummets to ~13,000 at 39 hours and 16,000 at 55 hours. In my opinion, these samples should not be grouped artificially and that the analysis of transcriptional profiles here will be driven more by time point 0.

Looking at the nMDS plot for these data, I would bet that in Fig 2a, third plot (Per-MAG normalized expression) the 3 separate dry communities at the bottom would belong to time point 0. I've included the figure in question (left) and added an edited version for visualization purposes (right). If my suspicions are correct then you may even observe a transition from dry to wet and back again.

Line 237: I got confused reading this with a higher relative abundance of rubryerthrin in 19 out of 20 MAGs. For clarity it would be useful to explain that this is 19 out of 20 MAGs that harbour this gene. Likewise for KatE with 18 out of 22 MAGs.

Line 249: How was it determined that Rubrobacteraceae01 was the most abundant MAG?

Line 299: The authors claim that light driven energy generation is stimulated minutes after rehydration (bacteriorhodopsin data), but their figure Fig. 3c, shows that rhodopsins are already transcribed at 0 hours. The extended data file even shows that there is no significant enrichment in transcriptional activity for these genes between 0 hours and 15 minutes in most cases.

Line 328: The authors claim that most populations oxidise PHA or use light in the early hydration phase. Yet only 4 MAGs used light for energy (I assume this refers to the bacteriorhodopsins) and only a few more (10) significantly used PHA at this stage. I wouldn't consider that to be most populations (out of 96).

Line 325: Bacteroidetes are now Bacteroidota.

Line 345: There is a follow up study to the Couradeau paper that may be a useful reference here: BONCAT-FACS-Seq reveals the active fraction of a biocrust community undergoing a wet-up event.

Lines 357: I was surprised not to see work such as "Soil microstructure as an under-explored feature of biological soil crust hydrological properties: Case study from the NW Negev Desert" considering the similarity of the sites and the relevance of hydration to the current study.

Line 390: The authors claim that certain enzymes are 'expressed' in dry and hydrated states. I think that 'transcribed' is a more accurate term here.

Figure 6. I'm not sure how the same photograph can be used to distinguish between Early hydration and Late hydration. Do you have any other photos of the biocrust? Moreover these images have been used in the Meier 2021 paper as well.

Minor corrections:

Line 58: "surviving rapid changes in osmotic changes" is a strange sentence.

Line 96: "und" should be "and"

Line 135: "singe" should be "single"

Line 319: I'm not sure what is meant by "uniformity" here.

Line 336: "this study changes" should be "this study challenges"

Line 560: "RNA-grad" should be "RNA-grade"

Line 610: "Brought 26% water content" should be "brought to 26% water content"

What is the stress values of the nMDS plots?

Reviewer #2 (Remarks to the Author):

Imminger et al. present a compelling single-cell activity study coupled to a metatranscriptomic approach that gives unique insight into microbial dynamics in biocrusts upon rewetting. The transcriptomic data paints a picture of microbes struggling under severe osmotic pressure while at the same time growing and metabolizing before desiccation sets in again. This is a valuable, tightly structured study with important implications for dryland soils and ecosystems. Below are suggestions that we hope will provide constructive feedback to finalize the manuscript. Note that this review greatly benefitted from the involvement of an ECR contributor. We are signing our names because we want the authors to know that if we appear blunt in our language, the idea is to be constructive. We encourage the authors to get in touch with us if there are any questions regarding our comments. We look forward to seeing this work in print.

Best regards,
Sebastian Kopf (sebastian.kopf@colorado.edu)
Tristan Caro (Tristan.caro@colorado.edu)

Major comments

Our major comment is in relation to the single cell growth rate data. Single cell growth data showing that filaments are all active (while single cells lag behind) is convincing. The authors have rigorously defined the assumptions/caveats going into their nanoSIMS analysis and engaged thoroughly with the literature on topics such as nanoSIMS dilution effects, assimilation constant from water (aw), and IMF. There has not yet been much published as to the dilution effects expected for D2O due to sample preparation, but the authors have addressed this confounding variable as best as possible. However, for full transparency to readers (especially those without prior experience with biological SIMS analyses), it would be good to directly address this effect in the main text by explaining that dilution of D2O via sample preparation is generally difficult to constrain, and that a correction is applied in order to account for this.

We consider this worth highlighting because it is conceivable that the relatively slow growth rates observed could result from an underestimate of the dilution factor (i.e., there could be more dilution during sample preparation than accounted for). Some of the presented data highlights this potential caveat. For example, in the photographs of Fig. 6, abundant growth and cell division of cyanobacteria is visible. One would expect that cyanobacteria that have visually doubled multiple times in the presence of isotopically labeled water would have more closely approached equilibrium with the label solution (30 at. %) rather than plateau around 4.5 at %. This plateauing of cyanobacterial growth, and the fact no cell reaches > 5 at %, could suggest that the cells have reached equilibrium with the isotopically labeled water, and that dilution effects during sample preparation are more substantial than accounted for. It is worth mentioning that the cell detachment solution with PVP((C6H9NO)_n) and Tween 20 (C58H114O26) could be playing a role in observed dilution, in addition to H-addition and solvent exchange from fixation, washing and staining. Because the effect of dilution can have such a profound effect on the measured growth rates, we suggest this caveat be discussed in the main text methods and discussion.

The transcriptomic data are very interesting. The authors help the reader digest this large and complicated dataset by separating it into distinct discussion sections, corresponding to different phases of the biocrust incubation, which is much appreciated.

In the "Microbial Populations generate energy ..." section (line 180), regarding the internal use of storage compounds – was the organic carbon content of the biocrusts measured or is it generally known that these samples are organic-poor? It would be good to clarify this in the manuscript to strengthen the discussion surrounding the PHA/glycogen degradation transcript abundance.

Are there any other physical/chemical characterization of the biocrusts available that could be included? These would be useful to mention/cite to help other drylands researchers better contextualize this system relative to others.

Line Comments

52: The authors define biomass "generation rate" in the manuscript and supplementary notes in units of inverse days [1/day], but report it here in days. Here, I think they mean generation "time" [days], the equivalent of growth rate but natural-log transformed to be more intuitive for the reader. It would be good to review this terminology throughout the manuscript to make sure biomass generation "rate" or "time" are used, respectively, when appropriate.

72: "burst of cells": it is unclear whether burst refers to a "burst of growth" as in rapid growth, or cells literally bursting due to osmotic stress. Specific terminology here could help clarify.

72: It is unclear why rapid rehydration and the loss of cells due to osmotic stress would necessitate rapid growth in a short time frame. What if these events lead to the loss of a microbiome steady state? And successive rewetting/drought events seriously impacted the function of the microbial community? Perhaps another way to phrase this would be as a hypothesis: you hypothesize that rapid growth does counteract the mass mortality that is observed upon rewetting.

81: "Desert bacteria" could be written as "bacteria found in desert ecosystems."

90: Consider rephrasing "the ideal model system". Biocrusts are certainly a strong model system, but perhaps "the ideal" is unnecessarily superlative. It would also be helpful for non-expert readers to include a sentence explaining how relevant/ representative biocrust systems are to drylands globally. Are most drylands devoid of plants and harboring high microbial biomass? How widespread are these kinds of biocrusts globally compared to other kinds of drylands? This would help better contextualize the system for readers who are unfamiliar with biocrusts.

112: Please state from the outset what the isotopic composition of your deuterated water label is.

295: Could clarify that "rapid resuscitation was detected" by transcriptomic data.

379: Would replace "repaired immediately" with "repaired rapidly".

418: How exactly was water added? Was it added dropwise, in order to preserve the physical structure of the biocrust? Or was a slurry generated?

Fig. 6: This figure was a bit difficult to parse at first. The comparison of the photos (top) to the main takeaways from the RNA-seq data is helpful. However, the dotted green line can be mistaken as the fraction of anabolically active cells, rather than the cell cartoons. What does the "ecosystem process" line map onto most directly?

Supplement

Eq. S2: Check typesetting - the formatting is a bit distorted.

Eq. S5: Check typesetting - formatting distorted a bit.

Line 26: Check typesetting - formatting of μ is distorted.

46: There is no citation 67 in the supplement. This should be corrected.

Reviewer #3 (Remarks to the Author):

Reviewer #1 comments:

Review for “Desert microbial communities are characterized by low productivity and ensure microbial-mediated processes by rapid resuscitation” by Imminger *et al.* submitted to Nature Communications.

General assessment

The manuscript presented by Imminger *et al.* documents the resuscitation of microbial community members from biological soil crust samples through the coupled use of Nano-SIMS and metatranscriptomics. The manuscript also relies on some of their previous data, notably metagenome-assembled genomes (MAGs), to consolidate their findings to show how nearly all community members activate immediately after a wetting event. The findings are interesting to a broad audience, the methods are robust, the writing clear and the figures are of high quality.

I have a few minor suggestions for the authors to clarify some of their results.

We appreciate the reviewer’s detailed look at our study and the general positive feedback. Many of the raised comments and suggestions are related to the omics methodology and its general abilities and limitations. For many of these topics we have conducted investigations and also collected experiences in other projects ourselves. Unfortunately, the necessity to present the main important findings to a wide audience in a concise format leaves little space for discussing these points in the main text of the manuscript. We therefore have added statements acknowledging the aspects you have highlighted and refer the readers to the supplementary materials (Supplementary Note 2) for more detailed descriptions and discussions.

Throughout our answers to the reviewer’s comments, the mentioned line numbers refer to the version of the manuscript in which all changes are depicted in the track changes mode.

Comments

Firstly, the catalogue of MAGs used from the previous work (Meier et al., 2021) comprises 96 populations. Checking the quality of these MAGs shows that just 2 can be considered high-quality genomes (primarily due to lacking ribosomal genes). In my opinion this is a shortcoming that should be mentioned or at least addressed by the authors. My concern is that the incompleteness of the genomes could obscure patterns relating to transcriptional activity. The average completeness of those genomes is ~83%, indicating that almost 1/5 genes could be missing from any population.

We fully agree with the reviewer and are aware of the noisiness of the data and thus have added a paragraph to the method section (lines 632-653) commenting on the purpose and advantages of transcriptome normalization, even with incomplete MAGs. Higher quality MAGs would result in even better transcription patterns and more genes passing the DeSeq2 significance thresholds. However, it is challenging to obtain high-quality MAGs from soil metagenomes. In this manuscript we are focussing only on the genes present in the MAGs and patterns that were detectable with statistical methods. While we would detect a larger catalogue of transcripts when looking at the bulk transcriptome data, it was difficult to see systematic transcription patterns, as the abundances of different populations vary between replicates and their transcriptional patterns are not fully synchronised. The main reason for using the per-MAG-normalised transcriptomes was to reduce the noisiness of the bulk data. Despite being incomplete (although 32 out of 96 have a >90% completeness), the MAGs enable a significant normalisation of the transcriptomic data when compared to the bulk, evident in the improved ordination clustering in Fig. 2a. One MAG may be 10 or 20% more complete than another, but their abundance might differ by 10x. Regarding transcription patterns, we are looking at transcript proportions that a population, a MAG, “assigns” to one or another metabolism. We notice when proportions of e.g. DNA repair transcripts and carbon acquisition-related transcripts assigned to the same MAG change. Lack of transcripts belonging to the 10-20% missing genes do not prevent us from detecting such changes. Also, we only report changes that were significant according to adjusted p-values of the relatively conservative DeSeq2 test (stated explicitly in lines 194-197; lines 658-666, marked in grey). A MAG with too few genes or transcripts or with seemingly random variation of transcript proportions due to important genes missing, would not show

many significant transcription changes in DeSeq2. While this might be the case for low abundant or incomplete MAGs, we are focussing on the many changes that are detected and on the fact that rapid transcription changes are happening across the vast majority of populations, indicating a reaction. Even with the conservative approach of adjusted DeSeq2 p-values, the majority of populations does not seem dormant.

Our study also shows that it is possible to perform such analyses in soil crusts, even with moderate metagenomic sequencing effort (5 samples). We hope that it will encourage other soil microbial ecologists to move beyond mere listing of transcribed genes and analysis of bulk data into analysis of transcription changes and mechanisms employed by different soil microorganisms. While we could detect significant patterns with our data, results of such studies will become even clearer with improvement of MAGs, e.g. through long-read technology.

Moreover, many of the MAGs capture the same diversity repeatedly. There are, for example, 15 members of the Rubrobacteraceae. If we expect that their genomes, and therefore their responses to hydration to be even moderately similar, it would inflate their specific response to hydration relative to, say, Microcoleus of which there is only a single representative.

This is precisely what our by-MAG-normalization approach tries to alleviate. We are not looking at a general transcriptomic response of all Rubrobacters, but at individual responses of various Rubrobacter MAGs. The Rubrobacteraceae MAGs could only be separated because they are different enough (probably different species). The strict transcript mapping at 99% identity ensures that we can differentiate between these related populations.

What inflates the weight of transcriptional response is not the number of different species but their relative abundance. For example, the bulk data patterns largely resemble the expression pattern of Rubrobacteraceae01 MAG and Microcoleus01 MAG as the abundance of their transcripts is much higher than that of other MAGs. Our normalization ensures that we can differentiate the expression patterns of other Rubrobacters from the abundant population (e.g. see different hydrogenase expression patterns in Fig. 4c).

Microcoleus is represented only by one MAG and its genome was not very abundant in the metagenome. However, one should keep in mind that it is a filamentous cyanobacterium with very large cells. All cells in the filament/bundle of filaments belong to one species = one MAG. Therefore, its biomass, albeit not necessarily DNA, dominates the crusts and its RNA dominates all transcriptome samples, and thus its transcription cannot be “overshadowed” even by 15 Rubrobacteraceae MAGs.

We paid specific attention to also analyze transcriptional responses of taxa represented by only a few or even one MAG in order to capture taxonomic and physiological diversity, e.g. Alphaproteobacteria, Bacteroidetes, Gemmatimonadaceae, and they are included in the figures. For example, the upregulation of DNA-repair and aerobic respiration genes at 15-30 minutes after rehydration can be observed for Gemmatimonadaceae01 and Cytophagales01 MAGs. The different phases of resuscitation can also be observed in the polysaccharide importer (SusCD-like genes) expression of Cytophagales01 MAG (see figure below).

Relative transcript abundances of genes encoding for polysaccharide uptake transporters (SusC- and SusD like proteins) across the temporal hydration phases for the Cytophagales01 MAG (Bacteroidota).

SusC- and SusD-like subunits encoded in two subsequent genes of the same contig are highlighted in bold font. The highest color intensity indicates the time point where the respective transcript reached its highest proportion in the MAG's transcriptome. This maximum value is indicated on the right in transcripts per million (grey bars). Note that Cytophagales01 expresses different SusC/D pairs, as well as different number of SusC/D pairs, dependent on the experimental phase. This might be a sign of utilization of different substrates during the resuscitation process and a switch to a wider variety of different polysaccharide substrates in the "main hydrated" phase.

Secondly, it wasn't until rereading the manuscript that I realized that the present study does not generate new MAGs given the wording "Genome-resolved metatranscriptomics" and "Population- resolved metatranscriptomics" is a bit ambiguous. Did the authors try to assemble and bin genomes from the metatranscriptomic data? The depth of the sequencing data suggests that this may have been possible (Extended Data Table 2).

We understand that the term genome-resolved metatranscriptomics might have caused confusion, since in genome-resolved metagenomics the genomes are produced from the metagenomic data itself. In this manuscript, we are using the MAGs published in a previous study (Meier et al. 2021) to resolve

transcriptomes stemming from exactly the same sample material from the same sampling event. Although both, metagenomes and metatranscriptomes investigations, were planned as a holistic study of the crusts, they had to be split into different manuscripts, due to the sheer amount of information.

We checked the manuscript text again to make sure that we sufficiently explained how the normalization was performed. In addition to the information in the method section (line 619) “The remaining reads were mapped to previously published metagenome contigs”), we now extended the introduction of our mapping reference in the results section to emphasise the use of previously generated metagenomic data (lines 160-165).

We realized, however, that we were using two terms (genome-resolved and population-resolved metatranscriptomic) when referring to the same: to transcriptomes that were mapped to previously obtained MAG, since for us MAGs represent population genomes. As this approach was previously called “genome-resolved metatranscriptomics” (e.g. <https://journals.asm.org/doi/10.1128/msystems.00474-21>, <https://doi.org/10.1111/1462-2920.14806>), we have now synchronized the two terms and only refer to “genome-resolved metatranscriptomics”.

We refrained from binning MAGs from transcriptomes as we see two shortcomings of this approach: 1) Only coding sequences are transcribed. Hence, transcriptome assembly results in short fragments with the length of a gene or at best an operon. Tetranucleotide frequencies would be variable across such short fragments and make binning by sequence composition challenging if not impossible. 2) Within a genome, expression varies between genes and expression of the same gene varies between timepoints. Some genes are expressed (i.e. present in the RNA data), others are not (missing from RNA data). From one time point to another, genes are upregulated by orders of magnitude. Others are downregulated. This makes binning by differential coverage, which assumes that fragments of one genome co-vary in their abundance (coverage) across samples, impossible. Furthermore, even if there would be a hypothetical way of attributing the different transcripts into a “MAG”, the fact that not all genes of a genome are expressed would lead to even less complete MAGs.

Transcriptome assemblies are used in order to improve functional annotation of transcripts as compared to annotation of short reads. Transcriptome assembly is only performed as a methodological shortcut when reference genomes are missing – either in pure culture studies where the presence of only one organism is certain and no binning is needed or in metatranscriptome studies where no reference metagenomes are available and only a bulk transcriptome analysis is intended.

Moreover, what proportion of the community do the authors estimate is captured by the 96 MAGs? It is claimed that this dataset encompasses all major taxa. For example, cyanobacteria comprise a very large proportion of the metatranscriptomic data but only constitute four MAGs by my understanding. It seems that ~20% of the transcripts mapped to ‘unbinned’ community members based on Extended Data Fig. 3. I wonder who these microbes are and how their absence from the mapping analysis amplifies the signals reported for the genomes captured as MAGs.

This is an important issue in environmental omics, which is unfortunately rarely addressed: what are the unbinned, unassembled, and unmapped fractions of omics data? We try to be transparent about it and report the read mapping percentages, both in the metagenomics manuscript, and the current transcriptomics-based study. The metagenomics manuscript contained estimates of the diversity covered by the sequencing effort based on raw reads. Furthermore, we provide comparisons of taxonomic composition between different analyses in both, metagenomic (Meier et al. 2021, Fig. 2) and current manuscript (Extended Fig. 3), although it should be noted that the databases used for different data (SILVA for 16S rRNA genes, GTDB for MAGs, and NCBI-Nr for taxonomic classification of contigs) differ in their size, taxonomic names, and precision of taxonomic classification.

We added a sentence to the results (lines 164-166) stating that our main analysis focuses on transcriptional responses of populations represented by the MAGs and referring the reader to now extended Supplementary Note 2 if the reader wishes to explore the relation between bulk data and the MAGs. The extended Supplementary Note 2 now contains information on the taxonomic composition of the bulk data and specifically comments on how bulk data is represented by the MAGs.

Knowing that we cannot capture the entire microbial community with the MAGs, it was important to us that the taxonomic and metabolic diversity of the community is reflected and key taxa are present. The only significant, although not very abundant, groups that we might be missing in the omics data are nitrifying archaea and spore-forming Firmicutes, that were observed at low abundances in 16S rRNA gene amplicon data (Meier et al. 2021). They are now mentioned in the Supplementary Note 2. As for cyanobacteria, please note that the filamentous species like *Microcoleus* sp. and *Coleofasciculaceae* have low diversity but high biomass. Therefore, they are only represented by one MAG each, but collect high numbers of transcripts, as evident in Source Data Table 3. It also can be seen in 16S rRNA amplicon data published in Meier et al. that *Microcoleus/Tychonema* is only represented by 2 OTUs, of which one makes up 18% of reads on average (up to 36%) and the other is marginal (0.1% on average, up to 1.3%). Of course, we see more very low abundant cyanobacterial taxa in the 16S rRNA data (both amplicon and metagenomic) such as e.g. *Nostocales* and some uncultured families. However, these taxa belong to the “long tail” of low abundant OTUs and we consider our MAGs representative of the cyanobacterial community structure with only few dominant members. Below is a figure of average relative abundances of cyanobacterial taxa found in the 16S rRNA gene amplicon data across 24 samples (Meier et al. 2021). Taxa represented by a MAG are marked with an arrow and corresponding MAG name.

The unbinned fraction consist of mostly short contigs, that were discarded from MAGs as “contamination”, were assigned to low quality MAGs that did not pass quality thresholds or were not assigned to any MAG or classified in any way. Over 60% are completely lacking hits to databases. The other represent largely the same taxa as represented by the MAGs (note that classification of short contigs based on protein-coding genes can be imprecise). These can be conserved genome regions shared between several related MAGs or fragments of lower abundant related populations.

The unbinned fraction of our dataset is rather small (25% of reads mapping to the assembly are assigned to unbinned contigs) compared to other metagenomic studies from complex environments. In the description of the binning tool DASTool (Sieber et al. 2018 <https://doi.org/10.1038/s41564-018-0171-1>), you can find data on how much diversity is covered by the MAGs when compared to the entire metagenome assembly (Sieber et al. 2018, Supplementary Table 3). In soil metagenomes analyzed by Sieber et al., this fraction ranges from 3 to 16%. For the biological soil crust metagenomes, the MAGs represent 24% - 36% (28% on average) of the raw metagenomic data (Meier et al. 2021, Table S2).

We added another supplementary figure (Supplementary Figure 6) summarizing the different fractions of the raw transcriptome data: assigned to MAGs, assigned to unbinned contigs, unmapped classified and unmapped unclassified (the largest) fractions. Accompanying text can be found in Supplementary Note 2.

I am not overly familiar with Nano-SIMS, however I would anticipate that non-filamentous cyanobacteria might exist in these Negev biocrust too. For example, one of the cyanobacterial MAGs is classified as a member of the *Thermosynechococcaceae* which are rod-shaped. Likewise, filamentous heterotrophs could be present in the biocrust samples (*Actinomycetota* for example).

Microscopically, cyanobacteria and heterotrophs can easily be distinguished by autofluorescence. Light emitted at 660 nm is the fluorescence of true chlorophyll, only present in cyanobacteria. In our microscopic analysis, we only observed filamentous cyanobacteria, identified by their autofluorescence. Therefore, we assume that any single-celled cyanobacterium such as Thermosynechococcaceae must have occurred at very low abundance. However, in our calculations of biomass generation and estimation of generation times based on ²H incorporation, we do not exclude the possibility for an autotrophic physiology of single cells. In order to capture any possible physiology (heterotrophic, mixotrophic and autotrophic) of the single cells, we estimated biomass generation considering a heterotrophic and autotrophic physiology, to capture all possible metabolisms of this cell type (Fig. 1).

We did, occasionally, observe small filaments without chlorophyll fluorescence (660 nm) in our microscopic investigations. As such we do not exclude the possibility for filamentous heterotrophs. However, as this was a very small fraction of cells, it was infeasible to target these as a separate group.

The estimate of doubling times is very interesting ecologically and has important repercussions for ecosystems processes. The doubling time estimates seem extremely high, or at least are not intuitive. The highest value of 471 days seems remarkably high to me, although this is an outlier. It's possible to estimate doubling time from the MAGs using codon usage bias estimates (e.g. gRodon: <https://github.com/jlw-ecoevo/gRodon2>) and I wonder how comparable these data would be with the high estimates presented here.

We agree that it would be interesting to use the metagenome data to infer doubling times of biocrust community members and have considered using gRodon in our study. However, we refrained from doing so, as we realised that gRodon is not really applicable to our data and questions, considering gRodon's scope and training model. The caution is extensively stated on gRodon's Github page (<https://github.com/jlw-ecoevo/gRodon2>): The tool was trained based on growth data of cultivated microorganisms growing in rich media under optimal growth conditions. Therefore, gRodon 1) estimates the maximum possible growth rate of a microorganism in pure culture under optimal conditions and 2) is bad at estimating growth rates of microorganisms which are underrepresented in culture collections. Therefore, gRodon can only estimate maximum growth rates of up to 5 hours doubling time. Everything above 5 hours is considered very slow growth and is just that: "above 5 hours". The training data of gRodon is thus very different from growth conditions of organisms in complex communities in the environment: conditions are far from optimal, far from stable, and the growth does not start from a small number of cells in a test tube with media, but happens in densely populated diverse communities. Furthermore, the microorganisms populating biological soil crusts have very few cultured representatives and the approach of gRodon was never tested for MAGs, only for genomes of cultures.

Line 119 - 121: p-values are reported yet the actual test performed is missing. Also, intriguingly, all p-values are identical ($p < 0.0013$) which seems at best unlikely and at worst suspicious. Please clarify these results and the test used to generate the statistics.

As described in the Methods section and the Supplementary Notes, we assessed the anabolic activity of single cells by consideration of (i) the measured deuterium content relative to the mean deuterium content detected in the control sample and (ii) the analytical uncertainty associated with NanoSIMS measurement data. In both cases we applied an approach frequently utilized in Analytical Chemistry, which states that a measurement signal is only associated with the presence of an analyte (in our case ²H) if it exceeds the mean plus three standard deviations registered on the blank (a sample which is identical with the measured

sample material but not containing the analyte, in our case the dead control). Given a Gaussian distribution of the measurement data, the probability for an erroneous detection of the analyte is less than 0.135%, corresponding to $p = 0.00135$ (value obtained from the Gaussian cumulative distribution function). This means that, if every data point is considered individually, it shows a p value of smaller than 0.00135.

Line 142: The authors state that their transcriptomics reveals reactivation of diverse microbial members after wetting. However when looking at the transcriptomic data in source data files it seems that more genes are transcribed across MAGs at 0 hours than most other time points when considering the sum of all genes transcribed per time point. This indicates that microbes are not dormant during dry condition (i.e. are actively transcribing) and that the reactivation of microbes is really a reactivation of specific pathways. Is my understanding correct here?

We apologise for this confusion and thank you for identifying this inconsistency. We went back to the R code used to generate the specific Table presented in the manuscript. While the code uploaded to the github repository was correct, counting all genes that have more than 0 TPMs in a MAG, the table uploaded with the manuscript was generated differently. While the intention was to count all genes with more than 3 reads mapping, considering only those as true positives, the R code wrongly applied this calculation to the TPM table (relative abundances, not counts), which is of course incorrect. We performed this calculation in a correct way and it shows (even clearer) the same trend as counting of non-zero TPM genes: The number of transcribed genes increases with hydration and falls again after desiccation. We updated the Source Data Table 2 accordingly.

Now the highest number of transcribed genes correctly occur at 15 min (transition from dry to hydrated) and at 6 and 12 h. Please note that the number of transcribed genes detected by transcriptomics depends on the evenness of distribution of transcripts between the different genes and on sequencing depth. Indeed, we look at reactivation of specific pathways (e.g. respiration, DNA repair, acquisition of organic carbon), at change of expressed functions between dry and hydrated, but not necessarily at the absolute numbers of detected expressed functions.

Regarding transcripts in dry samples, we do not consider the presence of RNA per se to be a sign of activity, or more precisely active transcription. RNA can be retrieved from ancient dry plant seeds, and we think that RNA persists in desiccated but e.g. trehalose-preserved cells. For most MAGs the number of transcribed genes increases with rehydration and also overall the diversity and evenness of transcripts in the community increases.

Line 230: The grouping of samples from time points 0, 39 and 55 hours after wetting as 'dry phase' is slightly odd. Clearly the 39 and 55 hours samples had been wetted and had subsequently dried out, however it is unknown (I assume) as to how long the time point 0 had been dry. I think the authors are grouping 'recently dried biocrust' with a 'control' prior to wetting. Another reason why this grouping doesn't please me is because there are large discrepancies in the number of genes transcribed at these different times. Taking the average of the triplicate sequencing, the time 0 has ~25,000 genes transcribed across all MAGs while this value plummets to ~13,000 at 39 hours and 16,000 at 55 hours. In my opinion, these samples should not be grouped artificially and that the analysis of transcriptional profiles here will be driven more by time point 0.

Looking at the nMDS plot for these data, I would bet that in Fig 2a, third plot (Per-MAG normalized expression) the 3 separate dry communities at the bottom would belong to time point 0. I've included the figure in question (left) and added an edited version for visualization purposes (right). If my suspicions are correct then you may even observe a transition from dry to wet and back again.

You are correct in seeing that the samples that have dried out again are different from the timepoint 0 (T0 shown light green in your modified figure), which has been dry for 8-9 months (dry in the field, according to weather data, plus during storage). (After correcting the calculation of numbers of expressed genes per timepoint as described in the comment before, the difference between T0 and T39 & T55 is smaller, though it still exists.)

Our differential expression analysis not only compares big clusters of samples such as “dry” vs “hydrated”, but primarily compares gene expression between subsequent time points. The additional comparison between all dry and all hydrated samples offers statistical power to detect very general basic differences in gene expression between these two conditions. To be statistically significant, such general differences in gene expression have to “stand out” above the variation within these large groups.

While the dry samples at the end of the experiment differ from the dry samples at the beginning of the experiment, the dry samples are still more similar to each other than to hydrated samples (see highly significant ANOSIM values in Fig. 2a and cluster dendrogram in Extended Fig. 4), which is why we decided to group them in one big “dry phase” cluster. The ANOSIM values in Fig. 2a indicate that differences between clusters as they are shown on the NMDS are much higher than differences within clusters. This is thus not an artificial grouping, but a grouping based on hierarchical clustering and confirmed by ANOSIM. Please note that the x-axis of the NMDS figure covers a wider range of dissimilarity than the y-axis. If scaled equally, the plot would look as follows:

However, we agree that the dry samples could be further subdivided in T0 and time points after the rehydration experiment. We explain the reasons for the difference between timepoint 0 and 39 and 55 hours as follows: With all our effort to keep the conditions close to natural, there still will be differences compared to a real desert rain event. For instance, in the climate chamber we cannot reach the incredible light intensities of real desert, we are lacking shrub plants and fauna (e.g. snails and arthropods) in our experiment, and the desiccation most likely proceeds faster without the buffer of surrounding soil. In this respect, it is remarkable that the transcriptional profiles of timepoint 55 do resemble the timepoint 0, even if they are not exactly the same.

Since we agree that the observed difference among the dry samples is worth mentioning, we have now marked the T0 samples in Fig. 2a and its corresponding legend. Further, we added a note acknowledging this difference in results (lines 180-183) and in the Supplementary Note 2.

Line 237: I got confused reading this with a higher relative abundance of rubryerthrin in 19 out of 20 MAGs. For clarity it would be useful to explain that this is 19 out of 20 MAGs that harbour this gene. Likewise for KatE with 18 out of 22 MAGs.

Clarifications have been added (lines 258/259).

Line 249: How was it determined that Rubrobacteraceae01 was the most abundant MAG?

Clarification and reference have been added (line 270): the MAG was most abundant in the metagenomic data based on read coverage. In transcriptomic data it collects the third-highest number of transcripts after *Microcoleus01* and *Coleofasciculaceae01* (both filamentous cyanobacteria).

Line 299: The authors claim that light driven energy generation is stimulated minutes after rehydration (bacteriorhodopsin data), but their figure Fig. 3c, shows that rhodopsins are already transcribed at 0 hours. The extended data file even shows that there is no significant enrichment in transcriptional activity for these genes between 0 hours and 15 minutes in most cases.

Thank you for pointing out this inconsistency between results and discussion. The sentence in the discussion meant to highlight sources of energy available in the early hydrated phase. But to be correct, only the respiration of storage compounds is “stimulated”, the rhodopsins and hydrogenase transcripts are already present in the dry state. We modified the sentence (lines 321-325).

Line 328: The authors claim that most populations oxidise PHA or use light in the early hydration phase. Yet only 4 MAGs used light for energy (I assume this refers to the bacteriorhodopsins) and only a few more (10) significantly used PHA at this stage. I wouldn’t consider that to be most populations (out of 96).

You are correct that they are ‘many’ not ‘most’. We changed the phrasing (line 354). However, we are referring here to more than 14 MAGs. The phrase does not only refer to what was significantly higher in “early hydrated” than in any other phase, but also to MAGs that had these transcripts already in the “dry phase” and to MAGs that continued expressing them in the “main hydrated phase”.

Line 325: Bacteroidetes are now Bacteroidota.

Thank you for spotting this. It has been corrected.

Line 345: There is a follow up study to the Couradeau paper that may be a useful reference here: BONCAT-FACS-Seq reveals the active fraction of a biocrust community undergoing a wet-up event.

We are aware of the study from Trexler et al. 2023, which was published during the submission process of our manuscript. Both studies differ in their methodological approaches and definitions of early and late

wetting time points (further explained below). Therefore, we do not believe that a direct comparison between the two studies is easy to achieve and have thus refrained from including this reference.

In our metatranscriptome study we investigated the response of bacteria to wetting within minutes, as our earlier timepoints are 15 and 30 minutes. We define this as the “early rehydration”, while the early timepoint in Trexler et al. is after 4 hours. In our study, comparable timepoints (from 3 hours on) define the “late hydration state”. This is surely attributed to the different methodologies used (BONCAT versus metatranscriptomics) but does not allow comparing the “early” phases of both studies directly.

Regarding methodological approaches, both studies use intrinsically different principles to detect active cells, most likely associated with different sensitivities. Trexler et al. used the incorporation of the methionine homologue HPG during protein synthesis, while we detected changes in transcript abundances after a water pulse (metatranscriptome) or the incorporation of ^2H from heavy water ($^2\text{H}_2\text{O}$) in the nanoSIMS assay. All these methods will detect microbial activity differently – at the transcript level (metatranscriptome), at the step of translation (BONCAT) or e.g. via lipid synthesis ($^2\text{H}_2\text{O}$ -nanoSIMS). The finding by Trexler et al. of differentially abundant taxonomic groups between the “inactive cell fraction” and “active cell fractions” could also stem from a limitation in the use of HPG. Until now, it is unclear if all bacteria are able to take up this non-canonical amino acid. Trexler et al. further stated the possibility that bacteria could evade the assay by using “alternative sources of methionine in place of HPG”, as they noticed unexpected responses of Firmicutes. And additionally, a toxic effect of HPG has recently been discussed (Landor et al. 2023, <https://doi.org/10.1016/j.mimet.2023.106679>).

Further, the microbial community composition of the biocrusts investigated by Trexler et al. (from the Colorado Plateau, Utah) differs considerably from the community in biocrusts from the Negev Desert (Israel), which we investigated in our study. The crusts from the Colorado Plateau are dominated by Proteobacteria and contain high relative abundances of Firmicutes and Bacteroidota. In contrast, the crusts from the Negev Desert are dominated by members of the Actinobacteriota, Chloroflexota and Gemmatimonadota, the latter two largely missing in the Colorado Plateau crusts. Thus, differing reactivation patterns could also be explained by these differing microbial community compositions. Additionally, previous BONCAT-FACS based studies have reported a stark difference between the original microbial community and the one observed after the elaborate cell processing required for BONCAT analysis (Krukenberg et al, 2021, <https://doi.org/10.3389/fmicb.2021.763971>). As BONCAT-FACS involves a multi-step treatment (particle detachment, filtration, incubation, filter detachment, in-between storage) of unfixed cells, this can lead to cell damage and thus altered community composition. When optimizing our cell detachment protocol that involves cell fixation and detachment, we have demonstrated that the recovered microbial community is very close to the original soil community in its composition (Eichorst et al. 2015, <https://doi.org/10.1093/femsec/fiv106>).

Lines 357: I was surprised not to see work such as “Soil microstructure as an under-explored feature of biological soil crust hydrological properties: Case study from the NW Negev Desert” considering the similarity of the sites and the relevance of hydration to the current study.

We appreciate making us aware of this reference, which we have added now (together with another reference) in lines 342/343 to add further information on the crust hydrological properties.

Line 390: The authors claim that certain enzymes are ‘expressed’ in dry and hydrated states. I think that ‘transcribed’ is a more accurate term here.

You are correct that enzymes/proteins are translated, while genes are expressed or transcribed. We changed the phrase accordingly (lines 420/421).

Figure 6. I’m not sure how the same photograph can be used to distinguish between Early hydration and Late hydration. Do you have any other photos of the biocrust? Moreover these images have been used in the Meier 2021 paper as well.

Thank you for pointing this out. The “green” crusts were chosen to symbolise “active” state, although of course crusts only turned green after 1-3 hours. For clarification and to avoid copyright issues, we replaced the pictures with new pictures, now showing indeed dry crusts, crusts after 30 minutes of rewetting and crusts after 12 hours after rewetting.

Minor corrections:

Line 58: “surviving rapid changes in osmotic changes” is a strange sentence.

The sentence has been changed for clarity (line 59).

Line 96: “und” should be “and”

Has been corrected (line 101).

Line 135: “singe” should be “single”

Has been corrected (line 144).

Line 319: I’m not sure what is meant by “uniformity” here.

We have removed the word “uniformity” for clarity (line 345).

Line 336: “this study changes” should be “this study challenges”

Has been corrected (line 365).

Line 560: “RNA-grad” should be “RNA-grade”

Has been corrected (line 606).

Line 610: “Brought 26% water content” should be “brought to 26% water content”

Has been corrected (line 678).

What is the stress values of the nMDS plots?

The stress values were added to NMDS plots in Fig. 2.

Reviewer #2+3 comments:

Reviewer #2 (Remarks to the Author):

Imminger et al. present a compelling single-cell activity study coupled to a metatranscriptomic approach that gives unique insight into microbial dynamics in biocrusts upon rewetting. The transcriptomic data paints a picture of microbes struggling under severe osmotic pressure while at the same time growing and metabolizing before desiccation sets in again. This is a valuable, tightly structured study with important implications for dryland soils and ecosystems. Below are suggestions that we hope will provide constructive feedback to finalize the manuscript. Note that this review greatly benefitted from the involvement of an ECR contributor. We are signing our names because we want the authors to know that if we appear blunt in our language, the idea is to be constructive. We encourage the authors to get in touch with us if there are any questions regarding our comments. We look forward to seeing this work in print.

Best regards,

Sebastian Kopf (sebastian.kopf@colorado.edu)

Tristan Caro (Tristan.caro@colorado.edu)

*We thank the reviewers for their assessment and their overall positive feedback. **Throughout our answers to the reviewers' comments, the mentioned line numbers refer to the version of the manuscript in which all changes are depicted in the track changes mode.***

Major comments

Our major comment is in relation to the single cell growth rate data. Single cell growth data showing that filaments are all active (while single cells lag behind) is convincing. The authors have rigorously defined the assumptions/caveats going into their nanoSIMS analysis and engaged thoroughly with the literature on topics such as nanoSIMS dilution effects, assimilation constant from water (aw), and IMF. There has not yet been much published as to the dilution effects expected for D2O due to sample preparation, but the authors have addressed this confounding variable as best as possible. However, for full transparency to readers (especially those without prior experience with biological SIMS analyses), it would be good to directly address this effect in the main text by explaining that dilution of D2O via sample preparation is generally difficult to constrain, and that a correction is applied in order to account for this.

We thank the reviewers for their assessment and their overall positive feedback. In response to this comment, we have added a statement referring to the effect of sample preparation on the isotope content and the need to apply correction factors in the main manuscript in the Results (lines 134-137) and the Methods (lines 477-480), in addition to the Supplementary Notes 1.

We consider this worth highlighting because it is conceivable that the relatively slow growth rates observed could result from an underestimate of the dilution factor (i.e., there could be more dilution during sample preparation than accounted for).

*We appreciate this concern and in response we have tested the dilution effect of our sample preparation pipeline on two bacterial strains with importance for the sample type – a *Rubrobacter* strain (as *Rubrobacter* is among the most abundant taxa in the investigated crusts, a gram positive bacterium) and a *Sphingomonas* strain that we isolated from the crusts (representing an Alphaproteobacterium also abundant in the crusts, a gram negative bacterium). Although not asked for, we believe confirming the applied dilution effect experimentally would give our results additional strength.*

We have described the results of this additional experiment in detail in the Supplementary Note 1. In short, our measurements on unfixed cells, and cells that underwent fixation and the cell detachment and

concentration procedure, matched very well the effect that we estimated based on the limited information in the literature. Thus, this test further strengthens the results and conclusion of our study – the first reported growth measurements in dryland soil illustrating slow growth of this well-adapted microbial community.

Some of the presented data highlights this potential caveat. For example, in the photographs of Fig. 6, abundant growth and cell division of cyanobacteria is visible. One would expect that cyanobacteria that have visually doubled multiple times in the presence of isotopically labeled water would have more closely approached equilibrium with the label solution (30 at. %) rather than plateau around 4.5 at %. This plateauing of cyanobacterial growth, and the fact no cell reaches > 5 at %, could suggest that the cells have reached equilibrium with the isotopically labeled water, and that dilution effects during sample preparation are more substantial than accounted for.

Unfortunately, it seems that the pictures we included in Fig. 6 were misleading. This conceptual figure should summarise the processes that occur during rehydration. The “green” crusts were chosen to symbolise an “active” state, although our investigated crusts only turn green after 1-3 hours. In the first minutes to 1 h after rehydration, the crusts appear moist, but no greening is visible. We apologize for this oversight and have replaced the pictures with new pictures, now showing dry crusts, crusts after 30 minutes of rewetting and crusts after 12 hours after rewetting.

*However, we want to emphasize that this greening effect is not caused by growth, but instead either by migration of cyanobacteria or the recovery of their pigments. In crusts from Spain that were dominated by *Oscillatoria* cyanobacteria, cyanobacterial filaments migrated to the surface of the crust upon rehydration (<https://www.nature.com/articles/35096640>, <https://link.springer.com/article/10.1007/s00248-002-0107-3>) and millimetres deep back into the crust when the crusts dried out. Then, the surface of the crusts appears devoid of cyanobacteria again. This observed migration is caused by hydrotaxis and independent of growth. In *Microcoleus*-dominated crusts from Oman, the greening effect has been explained by the recovery of pigments (<https://doi.org/10.1371/journal.pone.0112372>), while growth of cyanobacteria was only detectable after 2 days.*

In conclusion, the greening that we observe in the crusts from the Negev Desert is not caused by growth, but instead either by migration of cyanobacteria or recovery of their pigments. Thus, this observation does not stand in contrast to the replication times that we estimated.

It is worth mentioning that the cell detachment solution with PVP((C₆H₉NO)_n) and Tween 20 (C₅₈H₁₁₄O₂₆) could be playing a role in observed dilution, in addition to H-addition and solvent exchange from fixation, washing and staining. Because the effect of dilution can have such a profound effect on the measured growth rates, we suggest this caveat be discussed in the main text methods and discussion.

We agree that sample preparation will influence the isotope content of cells and have added a clarifying statement in lines 134-137 and lines 477-480, in addition to the Supplementary Notes 1. And as explained above, we now also present experimental data confirming the correction factor used to account for the dilution.

The transcriptomic data are very interesting. The authors help the reader digest this large and complicated dataset by separating it into distinct discussion sections, corresponding to different phases of the biocrust incubation, which is much appreciated.

We thank the reviewers for their assessment of our explanation of the metatranscriptome data.

In the "Microbial Populations generate energy ..." section (line 180), regarding the internal use of storage compounds – was the organic carbon content of the biocrusts measured or is it generally known that these

samples are organic-poor? It would be good to clarify this in the manuscript to strengthen the discussion surrounding the PHA/glycogen degradation transcript abundance.

We have added information on the organic carbon content in these biocrusts (0.31-0.86 %, Yu et al. 2014, <https://doi.org/10.1007/s00374-013-0856-9>; 0.6% measured in our experiment) in lines 447/448, which demonstrates a low content compared to other topsoils. We also refer to these values in the discussion (lines 357-359).

Are there any other physical/chemical characterization of the biocrusts available that could be included? These would be useful to mention/cite to help other drylands researchers better contextualize this system relative to others.

We referenced publications that provided information on the physical/chemical characteristics of the parental soil and biocrusts. In the revised version of the manuscript, we have now also added a summary on the physical/chemical characteristics of the parental soil and biocrust in lines 444-448.

Line Comments

52: The authors define biomass "generation rate" in the manuscript and supplementary notes in units of inverse days [1/day], but report it here in days. Here, I think they mean generation "time" [days], the equivalent of growth rate but natural-log transformed to be more intuitive for the reader. It would be good to review this terminology throughout the manuscript to make sure biomass generation "rate" or "time" are used, respectively, when appropriate.

This mistake has been corrected throughout the manuscript.

72: "burst of cells": it is unclear whether burst refers to a "burst of growth" as in rapid growth, or cells literally bursting due to osmotic stress. Specific terminology here could help clarify.

We were referring to the physical burst of cells and have clarified the meaning in the revised version (lines 74/75).

72: It is unclear why rapid rehydration and the loss of cells due to osmotic stress would necessitate rapid growth in a short time frame. What if these events lead to the loss of a microbiome steady state? And successive rewetting/drought events seriously impacted the function of the microbial community? Perhaps another way to phrase this would be as a hypothesis: you hypothesize that rapid growth does counteract the mass mortality that is observed upon rewetting.

*In the literature the possibility of cell loss due to rewetting or drying of soil is discussed (<https://doi.org/10.1016/j.soilbio.2020.107819>); however, the effect of rain on cell integrity in desert biocrusts is still uncertain. As we state in the abstract, it has been proposed that rehydration via rain causes cell death due to osmotic stress. However, this proposition is mainly based on experiments with isolated strains that mostly were not derived from desert biocrusts (e.g. *Pseudomonas putida* (<https://doi.org/10.1111/j.1365-2958.2004.04008.x>), *Deinococcus radiodurans* (<https://doi.org/10.1128/jb.178.3.633-637.1996>), *Arthrobacter siccitolerans* (<https://doi.org/10.1099%2Fjjs.0.052902-0>), *Sphingomonas* (<https://doi.org/10.1089%2Fast.2018.1840>)). In such experiments, viability decreased considerably after exposing the cells to desiccation stress. However, in soil, substantial cell loss during desiccation and rehydration has not been confirmed (<https://doi.org/10.2136/sssaj2003.7980>, <https://doi.org/10.1016/j.soilbio.2020.108012>,*

<https://doi.org/10.1007/s10533-020-00645-y>) and thus this point remains an unresolved question. A question we want to address in this study indirectly by investigating the potential for growth in a rain event.

One can discuss three possible scenarios regarding the effect of rehydration in desert soils:

1: A decent fraction of cells physically burst upon rehydration, resulting in the loss of a major part of the population and thus loss of steady state. If a major fraction of the community grows rapidly within a few days of sufficient water content, this growth will result in a new steady state, and it will also lead to maintaining of the reported higher microbial biomass compared to the underlying soil (<https://doi.org/10.1007/s00248-003-1004-0>).

2: A decent fraction of cells burst upon rehydration, resulting in the loss of a major part of the population and thus loss of steady state. But only a minor fraction of the community grows rapidly enough to divide in the few days of sufficient water content. Cell loss will continue with each rain event and will ultimately lead to extinction of entire populations and loss of microbial biomass. However, microbial biomass in biocrusts is consistently very enriched compared to the underlying soil (<https://doi.org/10.1007/s00248-003-1004-0>); therefore this scenario does not seem very likely.

3: Cells are well adapted to fluctuating osmotic conditions, thus only a minor fraction of cells physically burst upon rehydration. As a result, the system is still in steady state. The overall slow growth of cells has no negative effect on the microbial diversity or biomass of the system.

With our study, we basically tested scenario 1 and 3. Based on the detected slow growth combined with the majority of rain phases being very short (1-2 days), we conclude that scenario 3 explains best the stable community composition and enriched microbial biomass found in these desert biocrusts.

81: "Desert bacteria" could be written as "bacteria found in desert ecosystems."

We have rephrased this sentence (line 83).

90: Consider rephrasing "the ideal model system". Biocrusts are certainly a strong model system, but perhaps "the ideal" is unnecessarily superlative.

We have changed "ideal model system" to "a suitable model system" (line 95).

It would also be helpful for non-expert readers to include a sentence explaining how relevant/representative biocrust systems are to drylands globally. Are most drylands devoid of plants and harboring high microbial biomass? How widespread are these kinds of biocrusts globally compared to other kinds of drylands? This would help better contextualize the system for readers who are unfamiliar with biocrusts.

In the revised version, we have included information on the relevance of biocrusts in arid ecosystems (lines 93/94).

112: Please state from the outset what the isotopic composition of your deuterated water label is.

The information was added (30% $^2\text{H}_2\text{O}$, line 118).

295: Could clarify that "rapid resuscitation was detected" by transcriptomic data.

We clarified as suggested (line 316).

379: Would replace "repaired immediately" with "repaired rapidly".

Has been changed as suggested (line 409).

418: How exactly was water added? Was it added dropwise, in order to preserve the physical structure of the biocrust? Or was a slurry generated?

The water was added dropwise to preserve the physical structure of the crust. This information was added in line 455/456. Also, water was only added to 75% of their maximum water-holding capacity (corresponding to 26% H₂O / wet weight of rehydrated crust piece).

Fig. 6: This figure was a bit difficult to parse at first. The comparison of the photos (top) to the main takeaways from the RNA-seq data is helpful. However, the dotted green line can be mistaken as the fraction of anabolically active cells, rather than the cell cartoons. What does the "ecosystem process" line map onto most directly?

We agree that the message of Fig. 6 was a bit difficult to grasp. For clarification, we have now more clearly separated the aspects of the figure relating to growth and microbial activity that drives ecosystem processes. Regarding these processes, we were referring to H₂ oxidation, respiration and photosynthesis, which will commence rapidly after hydration.

Supplement

Eq. S2: Check typesetting - the formatting is a bit distorted.

Eq. S5: Check typesetting - formatting distorted a bit.

Line 26: Check typesetting - formatting of μ is distorted.

The problems with typesetting have been resolved.

46: There is no citation 67 in the supplement. This should be corrected.

Has been corrected.

Reviewer #3 (Remarks to the Author):

Reviewer #1 (Remarks to the Author):

The authors have done well to address all my critiques of their manuscript.

Reviewer #2 (Remarks to the Author):

Great work! See attached pdf for details.

Summary

We appreciate the diligence Imminger et al. have displayed in responding to our comments. Below are our responses to select comments from the authors. Please consider comments not included below to have successfully addressed our concern or suggestion. We whole-heartedly recommend this work for publication and look forward to seeing it in print.

Best regards,
Sebastian Kopf (Sebastian.kopf@colorado.edu)
Tristan Caro (Tristan.caro@colorado.edu)

Comments

We thank the reviewers for their assessment and their overall positive feedback. In response to this comment, we have added a statement referring to the effect of sample preparation on the isotope content and the need to apply correction factors in the main manuscript in the Results (lines 134-137) and the Methods (lines 477-480), in addition to the Supplementary Notes 1... We appreciate this concern and in response we have tested the dilution effect of our sample preparation pipeline on two bacterial strains with importance for the sample type...

Our main concern was in regard to dilution effects related to hydrogen/deuterium in acidic positions and during sample fixation. We believe authors have adequately tested and defended their approach, accounted for sources of uncertainty, and stated their assumptions/caveats. We especially appreciate the additional culturing experiments (Supplemental Note 1) to explore dilution effects and understand that these measurements are not trivial to undertake. Future work could be undertaken to further explore deuterium dilution effects under various conditions, but this is far outside the scope of the presented manuscript. If the authors felt inclined, they could include a 1-2 sentence perspective on this topic addressing remaining knowledge gaps, but such a statement would be at their discretion, and is not necessary for publication.

*Unfortunately, it seems that the pictures we included in Fig. 6 were misleading. This conceptual figure should summarise the processes that occur during rehydration. The “green” crusts were chosen to symbolise an “active” state, although our investigated crusts only turn green after 1-3 hours. In the first minutes to 1 h after rehydration, the crusts appear moist, but no greening is visible. We apologize for this oversight and have replaced the pictures with new pictures, now showing dry crusts, crusts after 30 minutes of rewetting and crusts after 12 hours after rewetting. However, we want to emphasize that this greening effect is not caused by growth, but instead either by migration of cyanobacteria or the recovery of their pigments. In crusts from Spain that were dominated by *Oscillatoria* cyanobacteria, cyanobacterial filaments migrated to the surface of the crust upon rehydration (<https://www.nature.com/articles/35096640>, <https://link.springer.com/article/10.1007/s00248-002-0107-3>) and millimetres deep back into the crust when the crusts dried out. Then, the surface of the crusts appears devoid of cyanobacteria again. This observed migration is caused by hydrotaxis and independent of growth. In *Microcoleus*-dominated crusts from Oman, the greening effect has been explained by*

the recovery of pigments (<https://doi.org/10.1371/journal.pone.0112372>), while growth of cyanobacteria was only detectable after 2 days.

In conclusion, the greening that we observe in the crusts from the Negev Desert is not caused by growth, but instead either by migration of cyanobacteria or recovery of their pigments. Thus, this observation does not stand in contrast to the replication times that we estimated.

We appreciate the discussion on this. Clarifying the difference between the timeframes of when photos were taken and when SIP sub-samples were acquired is helpful. Additionally, we appreciate the clarification of the greening effect not being caused by growth. The caption to Figure 6 could benefit from adding some of what you have included here for the benefit of readers less familiar with biocrust ecosystems and hydrotaxis but is not necessary.

Reviewer #3 (Remarks to the Author):
